# TF-Restormer: Complex Spectral Prediction for Speech Restoration

## Abstract

Speech restoration in real-world conditions is challenging due to compounded distortions such as clipping, band-pass filtering, digital artifacts, noise, and reverberation, and low sampling rates. Existing systems, including vocoder-based approaches, often sacrifice signal fidelity, while diffusion models remain impractical for streaming. Moreover, most assume a fixed target sampling rate, requiring external resampling that leads to redundant computations. We present TF-Restormer, an encoder-decoder architecture that concentrates analysis on input-bandwidth with a time-frequency dual-path encoder and reconstructs missing high-frequency bands through a light decoder with frequency extension queries. It enables efficient and universal restoration across arbitrary input-output rates without redundant resampling. To support adversarial training across diverse rates, we introduce a shared sampling-frequency-independent (SFI) STFT discriminator. TF-Restormer further supports streaming with a causal time module, and improves robustness under extreme degradations by injecting spectral inductive bias into the frequency module. Finally, we propose a scaled log-spectral loss that stabilizes optimization under severe conditions while emphasizing well-predicted spectral details. As a single model across sampling rates, TF-Restormer consistently outperforms prior systems, achieving balanced gains in signal fidelity and perceptual quality, while its streaming mode maintains competitive performance for real-time use. Anonymous code and demos are available at https://tf-restormer.github.io/demo.

## 1 Introduction

Speech enhancement (Ephraim & Malah, 1984; Pascual et al., 2017) has historically progressed through isolated sub-tasks with dedicated models such as denoising (Hu et al., 2020; Ho et al., 2020), dereverberation (Han et al., 2015; Wang & Wang, 2020), declipping (Mack & Habets, 2019), and bandwidth extension or super-resolution (Liu et al., 2022a; Lee & Han, 2021). In real-world settings, however, multiple distortions often coincide and are further compounded by digital distortions including lossy codecs (e.g., MP3, Ogg). These factors obscure magnitude and phase in the signal, making coherent, faithful speech restoration substantially more difficult.

This growing complexity has prompted a shift towards *general speech restoration* using generative models to handle diverse distortions (Liu et al., 2022b; Serrà et al., 2022). Vocoder (Kumar et al., 2019; Kong et al., 2020a)-based approaches reconstruct waveforms from compressed representations (Liu et al., 2022b; Andreev et al., 2023; Babaev et al., 2024). While such approaches improve perceptual quality, they discard phase information and treat speech as a semantic abstraction rather than a physical signal, producing output signals that deviate significantly from the input. In contrast, waveform-based generative models (Oord et al., 2016; Serrà et al., 2022) including GAN- and diffusion-based methods operate directly on the waveform, avoiding the abstraction bottleneck. Diffusion models (Welker et al., 2022; Richter et al., 2023) directly generate waveforms or spectra with improved fidelity, but their temporal compression or iterative sampling precludes streaming.

A further limitation of existing approaches lies in their assumption of a fixed target sampling rate. Restoration models, as well as dedicated super-resolution methods (Kim et al., 2024; Lu et al., 2025), invariably begin by resampling the input to the target rate before processing. This design choice simplifies model training but introduces critical drawbacks: every input must be converted

Figure 1: **Overview of TF-Restormer.** The model adopts an encoder-decoder design: TF-Encoder performs input-bandwidth analysis through stacked time and frequency modules, while the TF-Decoder reconstructs missing high-frequency bands with lightweight processing and learnable extension queries.

to the highest rate regardless of its native bandwidth, leading to redundant computation from re-sampling. Moreover, supporting multiple output rates requires either training separate models or repeatedly upsampling and downsampling, both of which are inefficient and impractical in real-world applications. These limitations call for a universal framework that analyzes input bandwidth directly and flexibly synthesizes arbitrary output rates without external resampling.

On the other hand, for denoising and separation tasks, models have been developed primarily in the complex STFT domain (Choi et al., 2018; Hu et al., 2020). In particular, recent works introduced time-frequency (TF) dual-path models (Dang et al., 2022; Wang et al., 2023), alternating sequence modeling along time and frequency axes. This design has proved effective for enhancement and separation tasks by preserving the frequency structure, and it naturally supports sampling-frequency-independent (SFI) formulations (Paulus & Torcoli, 2022; Zhang et al., 2023), since frequency bins can be treated as sequences whose length scales with the input rate while maintaining a consistent STFT frame duration. However, despite this flexibility, existing models still assume matched input-output rates. Moreover, because TF dual-path models explicitly preserve fine spectral structure, their computational cost grows substantially with higher sampling rates, making it difficult to apply to super-resolution tasks directly.

These limitations suggest the need for a new design that retains the strengths of TF dual-path processing while overcoming the inefficiency of fixed input-output rates. To this end, we present TF-Restormer, an encoder-decoder architecture for robust speech restoration under diverse degradations. Inspired by masked autoencoders (MAE) (He et al., 2022), TF-Restormer concentrates heavy processing in a TF dual-path encoder that analyzes the input bandwidth, while a lightweight decoder reconstructs the missing high-frequency components through learnable *extension queries* (Figure 1) with *cross-self* attention mechanism (Gupta et al., 2023). This *asymmetric* design enables arbitrary input-output sampling rates without external resampling, which minimizes redundant computation.

In summary, our contributions are as follows:

- We design an *asymmetric encoder-decoder* framework based on a TF dual-path Transformer with SFI-STFT: the encoder focuses on the input bandwidth, while the lightweight decoder extends high frequencies via learnable queries with a cross-self mechanism. This design enables general speech restoration, including super-resolution across arbitrary input-output rates without external resampling, with *a shared SFI-STFT discriminator* that supports uni-fied adversarial training across diverse rates.

- We improve robustness and practicality of TF dual-path processing by enhancing the frequency module with a projection-based *spectral inductive bias* and extending the time module to a causal variant for *streaming*, ensuring stable operation under extreme degradations as well as real-time applicability.

- We propose the *scaled log-spectral loss* as an auxiliary spectral objective that replaces conventional $\ell_1/\ell_2$ spectral objectives and complements perceptual (Babaev et al., 2024) and adversarial training (Mao et al., 2017). By selectively emphasizing reliably predictable regions, this loss stabilizes optimization under severe distortions while mitigating oversmoothing.

## 2 RELATED WORK

**Vocoder-based restoration** Vocoder systems typically project speech into Mel features (Liu et al., 2022b; Babaev et al., 2024) or learned representations (Koizumi et al., 2023; Li et al., 2024) and synthesize waveforms with neural vocoders (Oord et al., 2016; Kumar et al., 2019; Kong et al., 2020a). They often achieve "studio-like" perceptual quality but reduce fidelity, since intermediate features serve as perceptual cues rather than physical signals. Most also rely on temporally compressed U-Net structures (Pascual et al., 2017; Stoller et al., 2018), limiting real-time use. In contrast, we directly predict complex STFTs within a TF dual-path structure, retaining both magnitude and phase while enabling a streaming variant with minimal changes. Recent studies (Kaneko et al., 2022; Lu et al., 2025) show that spectral prediction with adversarial training can rival vocoder-based method.

**Diffusion-based restoration** Another approach is diffusion-based generation from waveform (Kong et al., 2020b; Serrà et al., 2022; Welker et al., 2022; Scheibler et al., 2024) or STFT inputs (Lemercier et al., 2023; Richter et al., 2023). While effective in perceptual quality, inference remains costly: even fast samplers require multiple denoising steps, hindering real-time use. Moreover, while diffusion excels at sampling diverse modes of a distribution, according to Babaev et al. (2024), speech enhancement typically seeks the most likely clean realization consistent with the observation, not a diverse set of alternatives. As a result, diffusion introduces unnecessary complexity for restoration or enhancement task. Therefore, we target a main-mode estimate via single-pass complex spectral prediction augmented with an adversarial loss for perceptual sharpness.

**TF dual-path models** TF dual-path models (Dang et al., 2022) alternate sequence modeling along time and frequency, preserving spectral structure and supporting SFI designs (Zhang et al., 2023). They show strong performance in enhancement (Cao et al., 2022; Lu et al., 2023; Chao et al., 2024) and separation (Wang et al., 2023; Saijo et al., 2024; Shin et al., 2025), but prior models used identical block designs for time and frequency without considering domain-specific differences. Under challenging degradations, we address this with a projection-based frequency module that injects spectral inductive bias for more robust high-frequency recovery. In addition, while prior dual-path models assumed matched input-output rates, we extend them with an encoder-decoder structure that supports super-resolution beyond fixed-rate restoration.

**Audio super-resolution** Conventional super-resolution models (Liu et al., 2022a; Han & Lee, 2022; Kim et al., 2024; Lu et al., 2025) typically assume a fixed target rate, upsampling inputs before processing to fill missing bands. While effective, this introduces redundant computation and ties each model to a single output rate, requiring repeated downsampling of outputs for different targets. In contrast, our encoder-decoder framework confines heavy processing to the input bandwidth and restores missing high-frequency bands through lightweight extension queries, improving efficiency. Unlike prior work restricted to predetermined rates, our design naturally supports user-specified outputs through the TF dual-path backbone.

## 3 TF-RESTORMER

### 3.1 SFI INPUT-OUTPUT FORMULATION

As an SFI model (Paulus & Torcoli, 2022), TF-Restormer addresses arbitrary input sampling rates $f_E$ by constructing STFT with a constant frame duration (SFI-STFT). Unlike conventional SFI that assumes matched input-output rates, we introduce the first *decoupled formulation*, enabling inference at user-specified output rates $f_D$. Given an input $x \in \mathbb{R}^{1 \times N_E}$ with sampling rate $f_E$, its STFT is $\mathbf{X} \in \mathbb{R}^{F_E \times T \times 2}$, where $F_E$ and $T$ are the number of frequency bins and frames. TF-Restormer then predicts $\mathbf{Y} \in \mathbb{R}^{F_D \times T \times 2}$ corresponding to an output $y \in \mathbb{R}^{1 \times N_D}$ at sampling rate $f_D$, satisfying $f_E : f_D = (F_E - 1) : (F_D - 1)$ under the assumption of consistent frame duration. To ensure universal applicability across sampling rates, we adopt a 40 ms analysis window with a 20 ms hop. This choice is a common unit in speech analysis and, being integer multiples across typical rates, guarantees consistent STFT construction at $\{8, 16, 22.05, 24, 32, 44.1, 48\}$ kHz without requiring resampling. The maximum number of frequency bins $F_{\max}$ is 961 for $f_E = 48$kHz.

### 3.2 ANALYSIS ENCODER AND EXTENSION DECODER

**TF-encoder for input analysis** As illustrated in Figure 1 and 2, the TF-Restormer is constructed with TF-encoder and TF-decoder. The TF-encoder is responsible for analyzing the speech component from the input signals $\mathbf{X} \in \mathbb{R}^{F_E \times T \times 2}$. Before the TF-encoder, the input complex representation $\mathbf{X} \in \mathbb{R}^{F_E \times T \times 2}$ is first projected to $C_E$ dimension by 2d convolution (Conv2D) layer with kernel

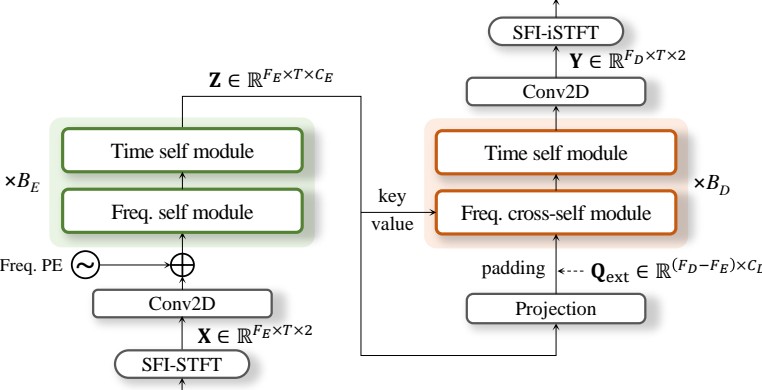

Figure 2: **Overall architecture of TF-Restormer.** With SFI-STFT and iSTFT, arbitrary input-output sample rates can be addressed in a single model. The encoder feature serves as both the input and the fixed key/value for the Freq. cross-self module in the TF-Decoder.

size of (3,3), followed by layer normalization (LN) (Ba et al., 2016). Then, sinusoidal positional embeddings are added along the frequency axis (Freq. PE). In the encoder, the projected feature is alternately processed by freq and time modules $B_E$ times to capture speech component.

**TF-decoder with extension query** Then, the encoder features $\mathbf{Z} \in \mathbb{R}^{F_E \times T \times C_E}$ are embedded to both the input of decoder by projection layer and key/value for cross-attention in the frequency cross-self module. To reconstruct the missing high-frequency region, we append the learnable extension-query to the projected encoder features $\mathbf{Q}_{\text{ext}} = [\mathbf{q}_{F_E+1}, \mathbf{q}_{F_E+2}, ..., \mathbf{q}_{F_D}]^T \in \mathbb{R}^{(F_D-F_E) \times C_D}$ where $\mathbf{q}_f \in \mathbb{R}^{C_D}$ is frequency-wise learnable query vector. They are sliced from a unified vector $\tilde{\mathbf{Q}}_{\text{ext}} = [\mathbf{q}_{F_{\min}}, \mathbf{q}_{F_{\min}+1}, ..., \mathbf{q}_{F_{\max}}]^T$ initialized over the frequency range $F_{\min} \leq F_E \leq F_D \leq F_{\max}$. Note that query values are shared across all the frames. In particular, the frequency module performs cross-attention based on extension query $\mathbf{Q}_{\text{ext}}$ as query and encoder feature as key/value. Then, the complex STFT values $\mathbf{Y} \in \mathbb{R}^{F_D \times T \times 2}$ are estimated from decoder features by Conv2D layer.

### 3.3 ASYMMETRIC TF DUAL-PATH MODULE

In TF dual-path modules, given the feature with shape $\mathbb{R}^{T \times F \times C}$ where $F \in \{F_E, F_D\}$, time modules process $F$ sequences with lengths of $T$ while frequency modules consider the feature as $T$ sequences with lengths of $F$ as illustrated in Figure 1. As a common structure borrowing from TF-Locoformer (Saijo et al., 2024) as shown in Figure 3, both time and frequency modules consist of two macaron-style (Lu* et al., 2019) convolution feed-forward network (ConvFFN) with Conv1D with kernel size $K$ for capturing local contexts. In ConvFFN, the expansion factor is 3 with SwiGLU as hidden activation. Between ConvFFN modules, multi-head self-attention (MHSA) is used for global contexts with $H$ heads. The time module performs MHSA on temporal frames with rotary positional encoding (RoPE) (Su et al., 2024) to offer the relative positions. On the other hand, the frequency module applies MHSA with the frequency projection layer to induce the structural bias as frequency bins are more static sequence with a fixed length, exhibiting a relatively consistent structural roles.

**Frequency cross-self module** For frequency cross-self module, we replace the first F-ConvFFN in the frequency self module with multi-head cross-attention (MHCA) based on key-value from the encoder feature $\mathbf{Z}_{\text{enc}}$ while query is high-frequency padded region by $\mathbf{q}_{\text{ext}}$, inspired by cross-self attention (Gupta et al., 2023). As illustrated in Figure 1, MHCA performs cross-attention based on extension query and key/value from encoder features. Therefore, MHCA conditionally operates when $f_E < f_D$ and extension query is padded, otherwise, bypassed.

**Attention with structural bias** Linformer (Wang et al., 2020) introduced linear projections of key-value to reduce the computations of attention, while MLP-Mixer (Tolstikhin et al., 2021) went further by replacing MHSA with static linear operations. Motivated by these insights, we incorporate a frequency linear projection (Fig. 3(c)) to impose an inductive bias for the structural consistency of frequency bins on top of the benefits of dynamic attention. Since frequency bins exhibit consistent characteristics, we share the same projection layer across all modules and key-value mappings. Formally, for each head $h = 1, ..., H$, given key-value $\mathbf{K}_{h,c}, \mathbf{V}_{h,c} \in \mathbb{R}^{T \times F}$ at channel $c$, a learnable projection matrix $\mathbf{A}_h \in \mathbb{R}^{F_{\max} \times F_{\text{proj}}}$ with dimension $F_{\text{proj}}$ and maximum bins $F_{\max}$ is applied as

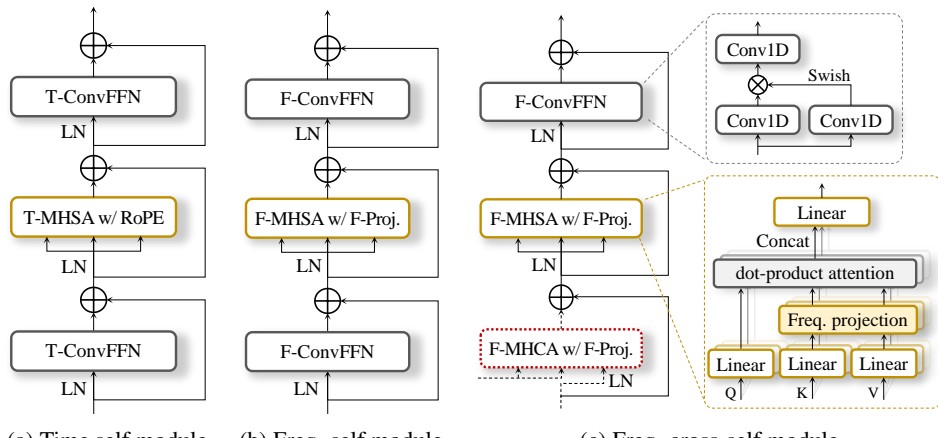

(a) Time self module    (b) Freq. self module    (c) Freq. cross-self module

Figure 3: **Unit modules in TF-Encoder and TF-Decoder.** The (a) time module is based on MHSA with RoPE while (b) the frequency encoder module is based on MHSA with frequency projection layer. (c) The frequency decoder module employs MHCA based on key/value from the encoder features.

$$\tilde{\mathbf{K}}_{h,c} = [\mathbf{K}_{h,c}, \mathbf{O}]\mathbf{A}_h \in \mathbb{R}^{T \times F_{\text{proj}}}, \quad 1 \le c \le C_h, \quad (1)$$

$$\tilde{\mathbf{V}}_{h,c} = [\mathbf{V}_{h,c}, \mathbf{O}]\mathbf{A}_h \in \mathbb{R}^{T \times F_{\text{proj}}}, \quad 1 \le c \le C_h, \quad (2)$$

where $\mathbf{O} \in \mathbb{R}^{T \times (F_{\max} - F)}$ is a zero-padding matrix.

**Streaming mode with mamba** The modular design of TF dual-path model further enables a seamless extension to streaming mode by replacing the time module with Mamba (Gu & Dao, 2024) blocks. Refer to Appendix C for detailed model configurations.

## 4 TRAINING

The model is trained by two phases of pretraining and adversarial training. The model is trained with $f_D$ randomly selected from $\{16, 24, 44.1, 48\}$kHz at each step by downsampling target speech signals from VCTK dataset (Yamagishi et al., 2019). Based on speech sources, we simulated noisy reverberant signals by convolving the room impulse response (RIR) and noise samples from the DNS dataset (Reddy et al., 2020). We then applied various digital distortions including codecs and downsampled the signal to the sampling rates $f_E$ of 8k or 16kHz, which are common in practical restoration condition (see Appendix B.1 for details). Because extension query could be undertrained if distribution of the input and output sample rates is unbalanced in training, we investigate these issue in Appendix F.

### 4.1 PRETRAINING

**Perceptual loss** Following Babaev et al. (2024), we incorporate a self-supervised learning (SSL)-based perceptual loss to stabilize adversarial training and encourage human-aligned quality. Specifically, extracting features from a pretrained SSL model for both the enhanced and clean waveforms, we minimize the mean-squared-error between these representations:

$$\mathcal{L}_{\text{p}}(\theta) = \mathbb{E}_{m,n}\left[\,|\phi(g_\theta(x))_{m,n} - \phi(s)_{m,n}|^2\,\right], \quad (3)$$

given that $y = g_\theta(x)$ is output of restoration model $g_\theta(\cdot)$ with parameters of $\theta$. $\phi(\cdot)_{m,n}$ denotes the $m$-th element of $n$-th frame from its feature map. We utilize WavLM-conv (Chen et al., 2022b) as in the previous study (Babaev et al., 2024).

**Proposed scaled log-spectral loss** Because the perceptual loss is restricted to 16 kHz and it is beneficial to guide spectral details to complement the looseness of perceptual loss, a previous work adopted an $\ell_1$ distance on the magnitude spectrum (Babaev et al., 2024). However, because TF-Restormer operates directly on the complex spectrum, the model can be explicitly supervised on both real and imaginary components in addition to the magnitude. Therefore, when denoting the STFT of target signal $s$ by $S_{m,tf} = |S_{r,tf} + jS_{i,tf}|$ and that of model's predicted signal $g_\theta(x)$ by $Y_{m,tf} = |Y_{r,tf} + jY_{i,tf}|$, we can extend to the complex domain as $\mathcal{L}_{\ell_1}(\theta) = \sum_{c \in \mathcal{C}} \alpha_c \cdot \mathbb{E}_{t,f}[|Y_{c,tf} - S_{c,tf}|]$, where $\mathcal{C} = \{r, i, m\}$ denotes the component index set and $\alpha_c$ are component weights.

However, even with complex supervision, while some regions are relatively easy to predict and receive consistent gradients, severely degraded or missing high-frequency regions yield unstable

gradients and drive the model toward oversmoothing, or averaging, effects (Babaev et al., 2024). To address this, we propose a *scaled log-spectral loss* selectively emphasizing well-predicted regions while preventing poorly predicted regions from dominating:

$$\mathcal{L}_{\mathrm{s}}(\theta) = \sum_{c \in \mathcal{C}} \alpha_c \cdot \mathbb{E}_{t,f} \left[ w_{tf} \log \left( 1 + \frac{|Y_{c,tf} - S_{c,tf}|}{w_{tf}} \right) \right], \tag{4}$$

where $w_{tf}$ is scale factor that controls the relative scaling of gradient. The formulation $w \log(1 + d/w)$ ensures large gradients on smaller distance $d$ than $w$, preserving regions where phase and magnitude are already reconstructed well, while suppressing the influence of large deviations, avoiding the averaging common in $\ell_1$ or $\ell_2$ losses. For choosing the weight value $w_{tf}$, we observed that a distance $|Y_{c,tf} - S_{c,tf}|$ tends to be proportional to the source $|S_{m,tf}|$. Therefore, $w_{tf}$ is empirically set to $E[S_{m,tf}]$ by averaging over the frames. We use $\alpha_m = 0.6$, $\alpha_r = 0.2$, and $\alpha_i = 0.2$. Finally, when combined with the perceptual loss, our proposed objective becomes $\mathcal{L}_{\mathrm{pre}}(\theta) = \lambda_{\mathrm{p}} \mathcal{L}_{\mathrm{p}}(\theta) + \lambda_{\mathrm{s}} \mathcal{L}_{\mathrm{s}}(\theta)$ where $\lambda_*$ denote loss weighting factors. Therefore, perceptual loss mainly focuses on largely deviated component while the scaled log-spectral more on well-predicted component, making them complementary to each other. For pretraining, we train the TF-Restormer with $\lambda_{\mathrm{p}} = 100$ and $\lambda_{\mathrm{s}} = 1$.

### 4.2 Adversarial Training

After pretraining the generator with $\mathcal{L}_{\mathrm{pre}}$, we introduce an adversarial loss component to reduce the artifacts and predict severely distorted or missing components. For adversarial training, we attach multi-scale STFT discriminators (Défossez et al., 2023) as $i$-th discriminator of $\varphi_i$ and apply least square GAN (LS-GAN) loss (Mao et al., 2017). For generator, generator LS-GAN and feature-matching loss (Kumar et al., 2019) terms are added, respectively:

$$\begin{aligned} \mathcal{L}_{\mathrm{gen}}(\theta) &= \lambda_{\mathrm{g}} \mathcal{L}_{\mathrm{g}}(\theta) + \lambda_{\mathrm{fm}} \mathcal{L}_{\mathrm{fm}}(\theta) + \lambda_{\mathrm{p}} \mathcal{L}_{\mathrm{p}}(\theta) + \lambda_{\mathrm{s}} \mathcal{L}_{\mathrm{s}}(\theta) + \lambda_{\mathrm{hf}} \mathcal{L}_{\mathrm{hf}}(\theta), \tag{5} \\ \mathcal{L}_{\mathrm{disc}}(\varphi_i) &= \mathcal{L}_{\mathrm{d}}(\varphi_i), \quad i = 1, ..., I. \tag{6} \end{aligned}$$

where $\mathcal{L}_{\mathrm{hf}} = \mathcal{L}_{\mathrm{pesq}} + 10 \cdot \mathcal{L}_{\mathrm{utmos}}$ is additional human-feedback loss (Babaev et al., 2024) for aesthetic quality with differentiable PESQ loss and UTMOS loss (Saeki et al., 2022). We performed adversarial training using $\mathcal{L}_{\mathrm{gen}}(\theta)$ with $\lambda_{\mathrm{g}} = 0.005$, $\lambda_{\mathrm{fm}} = 0.1$, $\lambda_{\mathrm{p}} = 100$, $\lambda_{\mathrm{s}} = 1$, and $\lambda_{\mathrm{hf}} = 0.0001$. Notably, we assign small weights to $\mathcal{L}_{\mathrm{g}}$ and $\mathcal{L}_{\mathrm{hf}}$ to avoid excessive generation artifacts.

**Proposed multi-scale SFI-STFT discriminators** In conventional adversarial training, a dedicated generator for each target sampling rate is trained with a corresponding discriminator as well (Défossez et al., 2023; Babaev et al., 2024; Ju et al., 2024), which introduces implementation overhead from coordinating adversarial schedules depending on the output sample rates. For training of a single generator across diverse rates, we propose a discriminator based on SFI-STFT, which preserves a consistent physical frame duration across sampling rates. Implemented with strided Conv2D layers, STFT discriminator (Défossez et al., 2023) produces two-dimensional maps that provide local real/fake supervision in the time-frequency plane. This design maintains sensitivity to spectral structure while remaining agnostic to absolute frequency resolution, thereby supporting adversarial training across different rates without redundant resampling or multiple discriminators. We employ 5 SFI discriminators with STFT window durations of $\{20, 40, 60, 80, 100\}$ ms.

## 5 Evaluation

### 5.1 Test Dataset and Metrics

We evaluate a unifed TF-Restormer for various datasets to validate the robustness across heterogeneous distortion conditions and arbitrary input-output sample-rates. For denoising(DN) and speech super-resolution (SSR), we also reports dedicated version of TF-Restormer for fair comparison and training configuration for dedicated models are summarized in Appendix B.3.

**UNIVERSE data for general speech restoration (GSR)** As GSR model, we evaluate on 100 synthetic samples generated by UNIVERSE authors (Serrà et al., 2022) to ensure comparability to prior works in $f_E = f_D = 16$kHz setting. The dataset introduces diverse simulated degradations such as bandpass filtering, reverberation, codec compression, and transmission artifacts.

**VCTK-DEMAND for DN** We additionally evaluated the well-known Valentini denoising dataset (Valentini-Botinhao & others, 2017) for direct comparison with conventional enhancement models as speech enhancement benchmarking. The evaluation set (824 utterances) consists of noisy mixtures from two speakers under four SNR conditions (17.5, 12.5, 7.5, and 2.5 dB).

Table 1: Results on UNIVERSE data for GSR. [†]We utilized pretrained models from implementation code from UNIVERSE++ (Scheibler et al., 2024). [‡]The results are reported in the original paper (Babaev et al., 2024)

| Model | Signal fidelity | | | | Semantic fidelity | | Non-intrusive quality | | |
|---|---|---|---|---|---|---|---|---|---|
| | PESQ[↑] | SDR[↑] | LSD[↓] | MCD[↓] | sBERT[↑] | sTokDis[↑] | WVMOS[↑] | UTMOS[↑] | DNSMOS[↑] |
| Input | 1.55 | 5.58 | 1.89 | 10.21 | 0.84 | 0.69 | 1.76 | 2.19 | 2.23 |
| Ground Truth | 4.50 | $\infty$ | 0.00 | 0.00 | 1.00 | 1.00 | 4.28 | 4.26 | 3.33 |
| VoiceFixer | 1.77 | -5.68 | 1.49 | 10.50 | 0.84 | 0.71 | 3.28 | 2.83 | 2.99 |
| StoRM | 1.76 | 9.01 | 1.67 | 6.87 | 0.84 | 0.70 | 3.14 | 2.70 | 2.94 |
| UNIVERSE[†] | 1.74 | 7.73 | 1.92 | 6.25 | 0.79 | 0.67 | 2.95 | 2.64 | 2.73 |
| UNIVERSE++[†] | 1.80 | 8.42 | 1.76 | 5.96 | 0.81 | 0.69 | 3.19 | 2.71 | 2.82 |
| TF-Locoformer | 2.13 | **11.61** | 2.00 | 6.26 | 0.89 | 0.76 | 3.20 | 2.95 | 2.86 |
| FINALLY | - | - | - | - | - | - | 4.43[‡] | 4.21[‡] | 3.25[‡] |
| TF-Restormer | **2.30** | 11.12 | **1.45** | **5.08** | **0.91** | **0.80** | 4.34 | 4.08 | **3.25** |
| TF-Restormer-*streaming* | 2.00 | 8.89 | 1.47 | 6.01 | 0.87 | 0.74 | 3.93 | 3.77 | 3.14 |

**VCTK for SSR**  For SSR evaluation, we construct paired data by downsampling 48 kHz clean utterances from the VCTK-0.92 dataset (Yamagishi et al., 2019). Beyond the clean case, we also create noisy-distorted conditions by adding degradations such as noise, reverberation, band-pass filtering, and codec effects, enabling a comprehensive evaluation of GSR with SSR. Note that the training simulation follows a similar procedure, which may provide a slight advantage to our model.

For the evaluation, we adopt non-intrusive perceptual estimators for mean opinion score (MOS): DNSMOS (Reddy et al., 2022), UTMOS (Saeki et al., 2022), and WVMOS (Andreev et al., 2023) to assess the perceptual quality. We also employ perceptual evaluation of speech quality (PESQ) (Rix et al., 2001) to assess the perceptual signal fidelity of restored signal compared to the reference. As a complement of metrics based on lower band, we consider signal-to-distortion ratio (SDR) (Le Roux et al., 2019), log-spectral distance (LSD), mel-cepstral distortion (MCD) (Fukada et al., 1992), and NISQA (Mittag et al., 2021) to assess the full-band signal (44.1/48kHz). In addition, to evaluate the reference-aware speech generation quality by capturing semantic congruence, we report Speech-BERTScore(sBERT) and SpeechTokenDistance(sTokDis) (Saeki et al., 2024). We also confirm effectiveness on real recordings based on DNSMOS, UTMOS, and WVMOS.

## 5.2 COMPARISON WITH EXISTING MODELS

For the UNIVERSE dataset, we consider VoiceFixer (Liu et al., 2022b) as a Mel vocoder-based baseline, StoRM (Lemercier et al., 2023), UNIVERSE (Serrà et al., 2022), and UNIVERSE++ (Scheibler et al., 2024) as diffusion-based baselines, TF-Locoformer as a recent TF dual-path Transformer model, and FINALLY (Babaev et al., 2024) as a latest strong Mel-vocoder method. As shown in Table 1, VoiceFixer improves MOS but sacrifices fidelity due to its Mel representation, while FINALLY achieves the highest perceptual quality yet lacks signal fidelity, a trend confirmed in Table 2. Diffusion-based methods yield more balanced results by directly operating in the waveform or complex STFT. TF-Locoformer preserves signal-level fidelity but suffers from residual perceptual artifacts and failure to recover lost details and naturalness (MOS, LSD). In contrast, TF-Restormer provides consistent improvements under fidelity, semantics, and perceptual quality, with its streaming variant maintaining competitive effectiveness under causal constraints. This indicates its robustness across diverse degradations in a universal restoration setting. Note that all the compared models are offline methods.

Next, we evaluate TF-Restormer on the VCTK+DEMAND focusing on denoising. In Table 2, we compare against DB-AIAT (Yu et al., 2022), MP-SENet (Lu et al., 2023), and TF-Locoformer as dedicated denoising models, and VoiceFixer, UNIVERSE, and FINALLY as universal restoration baselines. Since the input speech is already well preserved and only corrupted by additive noise, it favors models that minimize unnecessary generation and faithfully retain the input signal. Accordingly, dedicated denoising models outperform universal restoration models in terms of signal fidelity, as they are optimized to suppress noise without altering intact regions. In contrast, restoration models risk degrading reliability by over-modifying clean inputs, making them less trustworthy for such simple cases. While not surpassing dedicated denoising models in raw signal metrics, TF-Restormer achieves more consistent semantic and perceptual gains, showing strong generalization despite being designed for universal restoration. We additionally include a dedicated TF-Restormer variant; as expected, this task-matched version achieves the higher signal-fidelity scores.

Table 2: Results on VCTK-DEMAND for denoising task.

| Model | Signal fidelity | | | | Semantic fidelity | | Non-intrusive quality | | |
|---|---|---|---|---|---|---|---|---|---|
| | PESQ$^\uparrow$ | SDR$^\uparrow$ | LSD$^\downarrow$ | MCD$^\downarrow$ | sBERT$^\uparrow$ | sTokDis$^\uparrow$ | WVMOS$^\uparrow$ | UTMOS$^\uparrow$ | DNSMOS$^\uparrow$ |
| Input | 1.98 | 8.56 | 1.27 | 5.40 | 0.91 | 0.82 | 3.01 | 2.90 | 2.45 |
| Ground Truth | 4.50 | $\infty$ | 0.00 | 0.00 | 1.00 | 1.00 | 4.52 | 4.07 | 3.16 |
| DB-AIAT | 3.27 | 21.30 | 0.90 | 1.77 | 0.95 | 0.87 | 4.39 | 3.83 | 3.13 |
| MP-SENet | 3.61 | 21.03 | 0.85 | 1.58 | 0.95 | 0.88 | 4.35 | 3.86 | 3.12 |
| TF-Locoformer | 3.30 | **23.82** | 0.92 | 3.58 | 0.95 | 0.87 | 4.66 | 3.93 | 3.20 |
| VoiceFixer | 2.40 | -1.12 | 0.97 | 7.40 | 0.90 | 0.81 | 4.15 | 3.50 | 3.08 |
| UNIVERSE | 2.84 | 18.77 | 1.17 | 2.20 | 0.92 | 0.83 | 4.32 | 3.75 | 3.03 |
| FINALLY | 2.94 | 4.60 | - | - | - | - | **4.87** | **4.32** | **3.22** |
| TF-Restormer | 3.41 | 19.45 | 0.75 | 1.54 | **0.95** | **0.88** | 4.75 | 4.14 | 3.14 |
| TF-Restormer-*streaming* | 2.89 | 16.43 | 0.85 | 2.16 | 0.93 | 0.84 | 4.56 | 4.05 | 3.09 |
| TF-Restormer (*dedicated*) | **3.63** | 22.81 | **0.73** | **1.49** | 0.95 | **0.89** | 4.68 | 4.04 | 3.13 |

Finally, we experiment on the SSR task in Table 3, using a single model that directly supports arbitrary output sampling rates. For clean cases, we compare against dedicated super-resolution models: NVSR (Liu et al., 2022a), Frepainter (Kim et al., 2024), and AP-BWE (Lu et al., 2025), as well as VoiceFixer as a universal restoration baseline. As in Table 2, since the low-band of the input speech remains intact, dedicated models that concentrate on reconstructing the upper bands are favored. Unlike conventional approaches that rely on fixed input-output rates and often require zero-padding or redundant resampling, TF-Restormer leverages extension queries to dynamically expand the spectrum. With this versatility, TF-Restormer shows stable performance comparable to the dedicated models, faithfully retaining clean low-frequency regions while effectively generating high-frequency components. When the training is optimally aligned with the conventional method, the dedicated version of the proposed model shows improved results. In addition, under noisy-distorted conditions, TF-Restormer simultaneously restores corrupted regions and reconstructs missing high bands, demonstrating robust generalization beyond pure super-resolution. Overall, these results suggest the advantage of our model as a universal restoration framework that achieves bandwidth extension without sacrificing signal fidelity or requiring explicit resampling.

Table 3: Results on VCTK for SSR under clean and noisy-distorted conditions. [†]The models require fixed output sampling rates $f'$, thus evaluated by upsampling the input of $f_E$ to $f' \geq f_D$ and downsampling the output back to the target rate $f_D$. [‡]Dedicated models trained specifically for $f_D = 16$kHz.

| Method | 8kHz → 16kHz | | | 8kHz → 24kHz | | | 8kHz → 44.1kHz | | | 16kHz → 48kHz | | |
|---|---|---|---|---|---|---|---|---|---|---|---|---|
| | LSD$^\downarrow$ | MCD$^\downarrow$ | NISQA$^\uparrow$ | LSD$^\downarrow$ | MCD$^\downarrow$ | NISQA$^\uparrow$ | LSD$^\downarrow$ | MCD$^\downarrow$ | NISQA$^\uparrow$ | LSD$^\downarrow$ | MCD$^\downarrow$ | NISQA$^\uparrow$ |
| | *clean (SSR only)* | | | | | | | | | | | |
| Input | 2.53 | 1.84 | 3.78 | 2.91 | 2.03 | 3.78 | 3.44 | 2.44 | 3.78 | 3.17 | 1.31 | 4.40 |
| NVSR[†] | 0.83 | 1.62 | 4.15 | 0.89 | 1.82 | 4.24 | 0.94 | 2.06 | 4.16 | - | - | - |
| Frepainter[†] | 1.33 | 1.63 | 3.94 | 1.40 | 1.97 | 3.79 | 1.37 | 2.37 | 3.71 | 1.31 | 1.43 | 4.01 |
| AP-BWE[†] | 0.90[‡] | 1.33[‡] | 4.20[‡] | 0.86 | 1.36 | 4.34 | 0.88 | 1.58 | 4.26 | 0.85 | 1.38 | 4.33 |
| VoiceFixer[†] | 1.05 | 6.78 | 4.20 | 1.05 | 6.49 | 4.27 | 1.06 | 6.11 | 4.21 | - | - | - |
| TF-Restormer | 0.89 | 1.29 | **4.53** | 0.95 | 1.48 | **4.61** | 1.01 | 1.74 | **4.54** | 0.97 | 1.28 | **4.62** |
| TF-Restormer (*dedicated*) | **0.81** | **1.14** | 4.42 | **0.82** | **1.31** | 4.58 | **0.82** | **1.43** | 4.40 | **0.81** | **1.26** | 4.57 |
| | *Noisy-distorted (GSR + SSR)* | | | | | | | | | | | |
| Input | 3.36 | 11.38 | 1.91 | 3.49 | 11.60 | 1.91 | 3.64 | 11.47 | 1.91 | 3.48 | 11.37 | 1.73 |
| VoiceFixer[†] | 1.36 | 7.62 | 3.73 | 1.35 | 7.32 | 3.91 | 1.40 | 6.96 | 3.80 | - | - | - |
| StoRM | 1.76 | 4.57 | 3.97 | - | - | - | - | - | - | - | - | - |
| UNIVERSE++ | 1.79 | 5.28 | 3.39 | - | - | - | - | - | - | - | - | - |
| TF-Restormer | **1.16** | **2.78** | **4.49** | **1.21** | **2.97** | **4.54** | **1.18** | **3.08** | **4.52** | **1.18** | **2.86** | **4.54** |
| TF-Restormer-*streaming* | 1.30 | 3.93 | 4.42 | 1.31 | 4.01 | 4.49 | 1.30 | 4.05 | 4.46 | 1.26 | 3.86 | 4.46 |

## 5.3 MOS Evaluation on Additional Datasets

**VoxCeleb** We evaluate TF-Restormer on 50 real-recorded utterances (Su et al., 2020) from VoxCeleb1 (Nagrani et al., 2017) and compared with conventional method including DE-MUCS (Défossez et al., 2019) and HiFi-GAN-2 (Su et al., 2021). As shown in Table 4a, TF-Restormer achieves perceptual MOS scores (UTMOS, WVMOS, DNSMOS) comparable to recent vocoder- and diffusion-based models. While FINALLY (Babaev et al., 2024) remains one of the strongest perceptual-quality systems, our unified architecture delivers similarly natural outputs despite not being specialized for perceptual enhancement, demonstrating competitive robustness on real speech recordings.

**LibriTTS** We further evaluate on the LibriTTS test_other set provided in the Miipher release (Koizumi et al., 2023). TF-Restormer again reaches perceptual quality on par with FINALLY

Table 4: Evaluation of non-intrusive MOS results on real-recorded data(VoxCeleb, LibriTTS-test_other) and URGENT 2025 blind testset.

(a) VoxCeleb(real data)

| Model | UTMOS | WVMOS | DNSMOS |
|---|---|---|---|
| Input | 2.76 | 2.90 | 2.72 |
| VoiceFixer | 2.60 | 2.79 | 3.08 |
| DEMUCS | 3.51 | 3.72 | 3.27 |
| StoRM | 3.29 | 3.54 | 3.17 |
| HiFi-GAN-2 | 3.67 | 3.96 | 3.32 |
| FINALLY | **4.05** | **3.98** | 3.31 |
| TF-Restormer | 3.98 | 3.82 | **3.34** |

(b) LibriTTS(test other)

| Model | UTMOS | WVMOS | DNSMOS |
|---|---|---|---|
| Input | 3.41 | 3.69 | 2.88 |
| Miipher | 3.95 | 4.15 | 2.99 |
| FINALLY | **4.18** | **4.28** | 3.15 |
| TF-Restormer | 4.16 | 4.22 | **3.18** |

(c) URGENT 2025 (blind test set)

| Model | UTMOS | NISQA | DNSMOS |
|---|---|---|---|
| Input | 1.55 | 1.58 | 1.90 |
| Bobbsun(R.1) | 2.09 | 3.22 | 2.88 |
| rc(R.2) | 2.03 | 2.92 | 2.83 |
| Xiaobin(R.3) | 2.16 | 3.24 | 2.92 |
| wataru9871(R.13) | 2.53 | 3.74 | 3.10 |
| LLaSE-G1 | 2.09 | 2.93 | 2.80 |
| UniSE | 2.85 | 3.72 | **3.17** |
| TF-Restormer | **3.37** | **4.37** | 3.13 |

and Miipher, showing that the model generalizes well across different corpora while retaining its key advantage of supporting arbitrary sampling-rate restoration within a single framework.

**URGENT 2025**   Finally, we report non-intrusive MOS metrics on the URGENT 2025 blind test set (Saijo et al., 2025) compared to particating teams and lastest models including LLaSE-G1 (Kang et al., 2025) and UniSE (Yan et al., 2025). The official ranking incorporates both non-intrusive and intrusive measures, and the latter tend to favor deterministic bandwidth-preserving approaches while penalizing generative or reconstructive models. As a result, the top-ranked systems (Sun et al., 2025; Chao et al., 2025; Rong et al., 2025) are predominantly deterministic enhancers and generally obtain lower perceptual MOS. In contrast, TF-Restormer achieves natural-sounding outputs with strong non-intrusive MOS scores, showing that the model maintains stable perceptual quality on the URGENT blind test set as well.

### 5.4 ABLATION STUDY

To validate the effects of the proposed methods, we conduct an ablation study on scaled log-spectral loss, decoder design, and frequency projection module.

**Effects of scaled log-spectral loss**   In Table 5, we first assess whether auxiliary spectral losses provide benefits. Using perceptual loss alone leads to less stable optimization, whereas adding any spectral term consistently improves performance, confirming the importance of spectral constraints. Among regression-based losses, the $\ell_1$ loss on complex STFT components (magnitude, real, imaginary) outperforms magnitude-only variants by better preserving signal fidelity. Replacing $\ell_1$ with $\ell_2$ slightly degrades performance, likely due to oversmoothing. These results indicate that perceptual loss is essential for high-level quality but must be paired with an appropriate spectral objective.

We next compare log- and scaled log-spectral formulations. A plain log1p loss behaves similarly to $\ell_1$ because typical spectral distances are far below 1, keeping its gradient near 1. The proposed scaled log-spectral loss provides additional gains by adjusting gradient magnitude according to the target spectrum: suitable scale values balance well-aligned and poorly aligned regions, whereas overly small scales collapse gradients and damage performance. Removing perceptual loss noticeably harms both $\ell_1$ and s-log1p, and in this setting $\ell_1$ remains more stable, showing that s-log1p is not effective as a standalone objective. The best overall results arise when $w_{tf}$ is adaptively derived from the target magnitude, demonstrating that the proposed magnitude-adaptive scaling offers the most reliable trade-off between fine spectral detail and global coherence.

Table 5: Ablation Study on spectral loss. log1p denotes $\log(1 + d)$ while s-log1p is the proposed scaled log1p $w \log(1 + d/w)$ where $d$ is $\ell_1$ distance.

| Type of spectral loss | Perceptual loss | UNIVERSE(GSR) | | | VCTK+DEMAND(DN) | | | VCTK(SSR,8→16kHz) | | |
|---|---|---|---|---|---|---|---|---|---|---|
| | | PESQ$^\uparrow$ | MCD$^\downarrow$ | UTMOS$^\uparrow$ | PESQ$^\uparrow$ | MCD$^\downarrow$ | UTMOS$^\uparrow$ | PESQ$^\uparrow$ | MCD$^\downarrow$ | UTMOS$^\uparrow$ |
| None | ✓ | 1.85 | 7.23 | 4.02 | 2.74 | 4.16 | 4.08 | 3.05 | 2.47 | 4.11 |
| $\ell_1$-norm (mag. only) | ✓ | 2.07 | 6.03 | **3.82** | 2.93 | 3.13 | **3.95** | 3.42 | 2.10 | **4.10** |
| $\ell_1$-norm | ✓ | **2.23** | **5.70** | 3.76 | 2.97 | 2.98 | 3.87 | **3.48** | **1.86** | 4.07 |
| $\ell_2$-norm | ✓ | 2.21 | 5.81 | 3.70 | 2.96 | **3.05** | 3.80 | 3.44 | 1.88 | 4.06 |
| $\ell_1$-norm | | 2.19 | 5.89 | 3.71 | **2.98** | 3.19 | 3.80 | 3.35 | 2.23 | 3.91 |
| log1p $(w_{tf}=1)$ | ✓ | 2.25 | 5.72 | 3.79 | 2.99 | 2.98 | 3.90 | 3.53 | 1.83 | 4.06 |
| s-log1p $(w_{tf}=10^{-3})$ | ✓ | **2.27** | **5.17** | **3.98** | **3.37** | **1.67** | 4.10 | **3.67** | **1.37** | 4.07 |
| s-log1p $(w_{tf}=10^{-4})$ | ✓ | 2.01 | 5.94 | 4.07 | 2.96 | 3.03 | **4.14** | 3.40 | 2.43 | **4.10** |
| s-log1p $(w_{tf}=10^{-3})$ | | 2.18 | 6.05 | 3.74 | 2.98 | 3.19 | 3.91 | 3.27 | 1.88 | 3.94 |
| s-log1p (adap. $w_{tf}$) | ✓ | **2.29** | **4.96** | **4.10** | **3.41** | **1.54** | **4.14** | **3.70** | **1.29** | **4.10** |

Table 6: Ablation Study. VCTK-ND denote noisy-distorted input from VCTK data in Table 3.

| STFT-Disc. | PESQ↑ | UTMOS↑ |
|---|---|---|
| **UNIVERSE** (*GSR*) | | |
| Separate | 2.27 | 4.08 |
| Shared (SFI) | **2.29** | **4.10** |
| **VCTK+DEMAND** (*DN*) | | |
| Separate | 3.28 | 4.07 |
| Shared (SFI) | **3.41** | **4.14** |
| **VCTK-C** (*SSR., 8 → 16*kHz) | | |
| Separate | 3.64 | 4.10 |
| Shared (SFI) | **3.70** | **4.11** |

(a) Effect of discriminator

| Case | Size(M) | MAC(G) | LSD↓ | MCD↓ |
|---|---|---|---|---|
| **VCTK** (*SSR, 8 → 16*kHz) | | | | |
| Enc-only. | 11.6 | 151.3 | 2.12 | 2.93 |
| w/o MHCA | 30.8 | 252.4 | 1.04 | 1.81 |
| w/ MHCA | 30.1 | 240.8 | **0.89** | **1.29** |
| w/ MHCA(S) | 10.9 | 89.2 | 1.36 | 2.29 |
| **VCTK** (*SSR, 8 → 44.1*kHz) | | | | |
| Enc-only. | 11.6 | 415.1 | 3.25 | 10.48 |
| w/o MHCA | 30.8 | 340.4 | 1.35 | **1.70** |
| w/ MHCA | 30.1 | 308.4 | **1.01** | 1.74 |
| w/ MHCA(S) | 10.9 | 156.8 | 1.44 | 2.11 |
| **VCTK-ND** (*SSR+GSR, 8 → 16*kHz) | | | | |
| Enc-only. | 11.6 | 151.3 | 2.23 | 4.39 |
| w/o MHCA | 30.8 | 252.4 | 1.20 | 3.30 |
| w/ MHCA | 30.1 | 240.8 | **1.16** | **2.78** |
| w/ MHCA(S) | 10.9 | 89.2 | 1.48 | 3.77 |

(b) Encoder-Decoder design

| Case | Size(M) | PESQ↑ | UTMOS↑ |
|---|---|---|---|
| **UNIVERSE** (*GSR*) | | | |
| w/o F-proj. | 28.1 | 2.26 | 3.90 |
| w/ F-proj.(separate) | 63.6 | **2.31** | **4.11** |
| w/ F-proj.(shared) | 30.1 | 2.29 | 4.10 |
| **VCTK+DEMAND** (*DN*) | | | |
| w/o F-proj. | 28.1 | 3.21 | 4.03 |
| w/ F-proj.(separate) | 63.6 | 3.38 | **4.15** |
| w/ F-proj.(shared) | 30.1 | **3.41** | 4.14 |
| **VCTK** (*SSR, 8 → 16*kHz) | | | |
| w/o F-proj. | 28.1 | 3.54 | 3.97 |
| w/ F-proj.(separate) | 63.6 | 3.545 | 3.94 |
| w/ F-proj.(shared) | 30.1 | **3.70** | **4.11** |

(c) Effects of frequency projection

**SFI-STFT discriminator** As shown in Table 10a, *a shared SFI*-STFT discriminator consistently outperforms *separate rate-specific* discriminators across all tasks. Because the SFI representation aligns TF structure across sampling rates, the unified discriminator receives more coherent supervision and produces more stable gradients. In contrast, separate discriminators see only partial bandwidth conditions, leading to weaker adversarial signals. These results confirm that the shared SFI design is more effective for multi-rate restoration.

**Encoder-Decoder design.** We further analyze the contribution of the encoder–decoder structure (See Appendix F for detailed illustration.). The *Enc-only* model removes the decoder entirely and applies nine encoder blocks after padding the extension queries at the input. Although this variant has a small parameter count (11.6M), its MACs are extremely large (151-415G), since the encoder must jointly infer the observed low band and synthesize the missing high-frequency components. The *w/o MHCA* variant restores the encoder–decoder structure but replaces the MHCA module with the same self-attention block used in the encoder, preventing the decoder from conditioning on encoder features. Our proposed *w/ MHCA* model incorporates cross–self frequency attention, enabling more reliable reconstruction through encoder-conditioned queries.

To further isolate the effect of architectural design from model size, we additionally include a reduced version of the proposed model (*w/ MHCA (S)*), whose parameter count matches that of the dec-only configuration. Despite having far fewer MACs than dec-only, this size-matched variant consistently outperforms the decoder-only model across all bandwidth settings (Table 6b), confirming that the gains arise from the explicit separation of analysis and reconstruction and the use of cross-attention rather than increased parameter count or computational cost.

**Effects of frequency projection.** Finally, we examine the influence of the *frequency-projection (F-proj.)* module. Introducing projection provides an explicit structural prior along the frequency axis, which stabilizes training and yields consistent improvements over the no-projection baseline across tasks. The difference between using *shared* and *separate* projections is relatively small, though the *separate* version exhibits less stable behavior in the super-resolution setting. More critically, the non-shared design is highly inefficient, expanding the model to 63.6M parameters. Given the similar performance and the large gap in model size, the shared frequency-projection module offers the most practical and efficient configuration.

# 6 CONCLUSION

We presented TF-Restormer, a speech restoration model with an encoder-decoder design that enables efficient operation across arbitrary input and output sampling rates. By concentrating on the input bandwidth with a strong TF dual-path encoder and extending high frequencies through lightweight decoder queries with a cross-self mechanism, TF-Restormer achieves balanced improvements in both signal- and semantic-level fidelity while also showing robust performance on single-task benchmarks such as denoising and bandwidth extension. We also proposed a shared SFI-STFT discriminator for unified adversarial training across diverse sampling rates. Finally, ablation studies confirm the effectiveness of our design choices, including the projection-based frequency module, the decoder design, and the proposed scaled log-spectral loss.

## REPRODUCIBILITY STATEMENT

We make our implementation code and demo anonymously accessible. Our anonymous link for pre-trained TF-Restormer and inference code is also included in demo pages: `https://tf-restormer.github.io/demo`

## ETHICS STATEMENT

We use only public speech corpora and collect no new personal data. We do not attempt speaker re-identification, and we do not redistribute raw audio. Aware of potential misuse (e.g., covert monitoring), we will apply access controls and intended-use restrictions and require legal compliance for any release.

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

## A    USE OF LARGE LANGUAGE MODELS

This paper was written by the authors. Large language models (LLMs) were used only for minor language polishing.

## B    DETAILS OF TRAINING PROCEDURE

### B.1    SIMULATION OF TRAINING DATASET

**Clean speech source**    The model is trained with VCTK training set. VCTK corpus (Yamagishi et al., 2019) is a multi-speaker English corpus containing 110 speakers with different accents. We split it into a training part VCTK-Train and a testing part VCTK-Test. The version of VCTK we used is 0.92. To follow the data preparation strategy of previous restoration studies Liu et al. (2022b), only the  *mic1* microphone data is used for experiments, and *p280* and *p315* are omitted for the technical issues. For the remaining 108 speakers, the last 8 speakers, *p360,p361,p362,p363,p364,p374,p376,s5* are split as test set VCTK-Test for super-resolution subtask. Within the other 100 speakers, *p232* and *p257* are also excluded because they are used in the test set VCTK-ND and VCTK+DEMAND datasets. Therefore, the remaining 98 speakers are used as training data.

**Simulation pipeline**    To simulate input signal for training, we randomly applied the various distortions based on the pipeline as shown in Figure 4. In particular, we sequentially applied physical and digital distortions. The physical distortions include convolution of transfer function mainly caused by reverberation at indoor environment. We used RIR samples from DNS dataset (Reddy et al., 2022). Note that we compensated time-delay effect from the convolution by applying direct component of RIR to the corresponding target speech signal. Then, as a second physical distortion, we added various background and interfering noises using noise samples (Reddy et al., 2022) and simulated colored gaussian noise. Each noise source is independently applied with signal-to-noise (SNR) ratio ranging from 0 to 20 dB. Then, as a final stage of physical distortion, we applied band pass filtering (BPF) to account for the recording condition of microphone such as occlusion, hardware properties, in this study, we mainly considered occlusion effect for the simulation. Also, to remove the phase distortion from the BPF, we applied as zero-phase filtering because the model does not need to consider these effect, only to make the learning process complicated. As a final step for physical simulation, we randomly scaled the level of signals from -35 to -15 dB Full Scale (dBFS). We also scaled the speech sources along with the corresponding input.

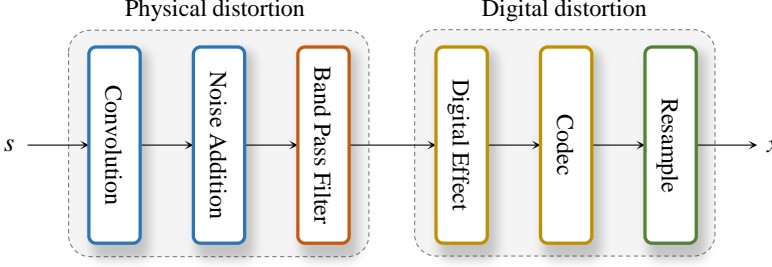

Figure 4: **Noisy-distorted speech input simulation pipeline.** The simulation procedure is partitioned to physical distortion and digital distortion.

Then, three kinds of digital distortions were simulated in sequence. We randomly applied audio clipping, crystalizer, flanger, and crusher as digital effects, each introducing characteristic nonlinear saturation, spectral over-enhancement, comb filtering, or quantization noise (detailed parameter ranges are summarized in Table 7). Afterward, digital codec compression was applied to emulate transmission artifacts, using either MP3 or OGG (Vorbis/Opus) encoding. Finally, the processed signals were randomly downsampled to 8 or 16 kHz to simulate low-bandwidth recording and communication scenarios.

Table 7: List of applied distortions with probabilities and parameter ranges.

| Augmentation | Prob. | Param. name | Range / Values | Notes |
|---|---|---|---|---|
| RIR convolution | 0.50 | - | - | direct-path delay compensated |
| Sample Noise | 1.00 | SNR (dB) | $[0, 20]$ | from DNS dataset |
| Colored Gaussian Noise | 1.00 | SNR (dB) | $[0, 20]$ | |
| | | exponent $\beta$ | $[0.75, 1.5]$ | |
| Band-limiting (BPF) | 0.5 | $f_1$ (Hz) | $[500, 1500]$ | zero-phase |
| (occlusion FIR) | | $f_2$ (Hz) | $f_1 + [200, 500]$ | transition band upper edge |
| | | cut_gain | $(0.1, 0.3)$ | stopband gain, applied as $g^\beta$ |
| | | $\beta$ | $[0.25, 1.00]$ | (thus effective stopband $\approx [0.22, 0.55]$) |
| | | taps | odd in $[31, 61]$ | `firwin2`, $f_s$=16k ( $f_N$=8k ) |
| Clipping | 0.5 | level (dB) | $[-15, 0]$ | hard clipping threshold |
| Crystalizer | 0.15 | intensity | $[1, 4]$ | spectral "sharpening" |
| Flanger | 0.05 | depth | $[1, 5]$ | short-delay comb filtering |
| Crusher (bit-depth) | 0.10 | bits | $[1, 9]$ | quantization/aliasing |
| Codec (any) | 0.30 | — | — | one of the following |
| MP3 | | bit rate (kbps) | $[4, 16]$ | variable bit-rate sampled uniformly |
| OGG | | encoder | vorbis, opus | random choice |
| Frequency Masking | 1.00 | $F_{bw}$ (freq. bins) | $[0, 10]$ | |
| | | # masks | $[0, 3]$ | set to $[0, 1]$ in adversarial training |
| Time Masking | 1.00 | $T_{dur}$ (frames) | $[0, 10]$ | |
| | | # masks | $[0, 2]$ | set to $[0, 1]$ in adversarial training |
| Downsample | 1.00 | target $f_s$ | $\{8k\ (0.25),\ 16k\ (0.75)\}$ | |

## B.2 TRAINING DETAILS FOR UNIFIED MODEL

For pretraining, TF-Restormer was optimized with a batch size of 2 on a single NVIDIA RTX 6000 Ada 48GB GPU using AdamW (Loshchilov & Hutter, 2019). Pretraining was run for 200,000 steps on the VCTK dataset with 3-second utterances. Adversarial training was then applied for an additional 200,000 steps. We used a learning rate of 2.0e-4 with betas (0.9, 0.995), applying a decay of 0.9 every 10,000 steps after 100,000 steps during pretraining, and every 10,000 steps during adversarial training.

Both stages used a 5,000-step linear warm-up for the generator. In adversarial training, the discriminator was updated twice per generator step (without warm-up), using AdamW with betas (0.8, 0.999). Following multi-scale STFT discriminator designs (Défossez et al., 2023), we employed our multi-scale SFI-STFT discriminator with STFT window sizes of [20, 40, 60, 80, 100] ms to capture spectral details at multiple resolutions.

Across all ablation variants, validation loss plateaued around 60k–70k steps, and no architecture exhibited signs of overfitting. We observed that checkpoint selection within this plateau region led to negligible performance differences, indicating that the chosen training length is sufficient for convergence and provides a fair comparison across variants.

## B.3 TRAINING DETAILS FOR DEDICATED MODEL

**VCTK+DEMAND** Since noise reduction does not require generating new speech components, prior work has shown that standard supervised learning is often sufficient. Therefore, in the fine-tuning stage we use small weights in adversarial loss ($\lambda_g = 0.001$ and $\lambda_{fm} = 0.01$ and the human-feedback perceptual loss ($\lambda_{hf} = 10^{-5}$). The model is trained to perform pure denoising following the standard VCTK+DEMAND training partition. The input and output sampling rates are both fixed to 16 kHz, and thus no extension queries are used in this setting.

**VCTK for Super-resolution** For super-resolution, the model is trained under the same protocol as conventional SR systems, using clean low-band inputs as supervision. Consequently, during adversarial fine-tuning the human-feedback loss is again applied with a small weight ($\lambda_{hf} = 10^{-5}$), as the task primarily focuses on recovering missing high-frequency content.

Table 8: Comparison of the model size and RTF. RTF is calculated on NVIDIA RTX 4090. [†]We utilized pretrained models from open implementation code from UNIVERSE++ (Scheibler et al., 2024). [‡]The model size of FINALLY includes WavLM whose model size is 358M.

| Model | Model Size (M) | $f_E \to f_D$ (kHz) | MACs(G) | RTF |
|---|---|---|---|---|
| VoiceFixer | 70.3 | $44.1 \to 44.1$ | 12.9 | 0.010 |
| StoRM | 55.1 | $16 \to 16$ | 156.4 | 0.520 |
| UNIVERSE[†] | 46.4 | $16 \to 16$ | 36.9 | 0.014 |
| UNIVERSE++[†] | 84.2 | $16 \to 16$ | 36.9 | 0.015 |
| FINALLY[‡] | 454.0 | $16 \to 48$ | – | – |
| TF-Locoformer | 14.9 | $16 \to 16$ | 246.9 | 0.025 |
|  |  | $48 \to 48$ | 731.6 | 0.088 |
| TF-Restormer | 30.1 | $8 \to 16$ | 240.8 | 0.009 |
|  |  | $8 \to 44.1$ | 308.4 | 0.017 |
|  |  | $16 \to 16$ | 440.9 | 0.034 |
|  |  | $16 \to 48$ | 518.7 | 0.053 |
| TF-Restormer-*streaming* | 19.0 | $8 \to 16$ | 114.7 | 0.012 |
|  |  | $8 \to 44.1$ | 138.1 | 0.018 |
|  |  | $16 \to 16$ | 214.5 | 0.035 |
|  |  | $16 \to 48$ | 242.0 | 0.049 |

## C  DETAILS OF MODEL CONFIGURATION

For TF-Restormer, $C_E$ and $B_E$ for encoder are set to 128 and 6 while $C_D$ and $B_D$ are set to 64 and 3. The kernel size in ConvFFN and the number of heads in MHSA/MHCA are commonly set to $K = 7$ and $H = 4$, respectively. For frequency projection layer, $F_{\text{proj}}$ is set to 512.

For offline TF-Restormer, each input mixture is normalized by dividing it by its standard deviation and the enhanced output is rescaled by the same factor. For streaming version of TF-Restormer, two mamba blocks are used in the time module with $d_{\text{state}} = 16$, causal Conv1D kernel size 3 with expansion factor 4. For streaming version, we still use the non-causal Conv2D layer for input and output projection for robust restoration, therefore the latency increases by two frames, total latency of 80ms (40 ms window, 20 ms hop). Overall, the model size of TF-Restormer is 30.1M for offline mode and 19.0M for streaming mode, which are smaller sizes compared to the existing models.

### C.1  COMPARISON OF THE MODEL SIZE AND RTF

In Table 8, we compare model size and multiply-accumulate operations (MACs) for a 1-second-long input using *ptflops* package[1]. We also measure real-time factor (RTF) measured on 4-second-long samples with an NVIDIA RTX 4090. Conventional models operate at fixed input-output sampling rates, which results in fixed MACs regardless of the task configuration. In contrast, TF-Restormer adapts its computation depending on the input and output rates $f_E$ and $f_D$.

Among baselines, StoRM requires 50 diffusion steps, leading to very high MACs and RTF despite its moderate model size. UNIVERSE and UNIVERSE++ reduce the number of steps (8 by default in the open implementation), which lowers the runtime cost compared to StoRM, but their model sizes remain relatively large and the diffusion process cannot be adapted for streaming, representing a fundamental limitation. TF-Locoformer, built on a dual-path design, involves higher computational complexity but benefits from effective parallelism, so its RTF is not as large as its MACs might suggest; its parameter size is also smaller than most diffusion- or vocoder-based systems.

Our proposed TF-Restormer also follows a dual-path formulation, so the raw MACs are relatively large. Nevertheless, RTF remains low in practice, comparable to or even faster than prior dual-path models. Crucially, TF-Restormer optimizes computation according to the input and output sampling rates: for instance, in the $8 \to 16$ kHz setting, redundant high-frequency processing is skipped, yielding a very low RTF. Also, the streaming variant maintains consistently low RTF while preserving accuracy, demonstrating its suitability for real-time applications.

---

[1]https://github.com/sovrasov/flops-counter.pytorch

# D  SIMULATION OF VCTK NOISY DISTORTED INPUT

The noisy-distorted input from VCTK testset in Table 3 was generated by corrupting clean VCTK utterances with additive noise from DEMAND (Thiemann et al., 2013) and colored Gaussian noise, RIR samples from RWCP (Nakamura et al., 2000) and AIR (Jeub et al., 2009) for reverberation, and distortions such as clipping and band-limiting. Additional digital effects including audio codecs (MP3, OGG) were applied before resampling to various rates (8-48 kHz). This simulation aligns the training pipeline while maintaining samples partitioning of speech, noise, and RIR sources. As a result, the average SDR of input data (in case of $f_E =$16kHz) is 2.11dB from 2937 utterances. 998 utterances (about 34%) are below SDR=0 dB and 234 utterances (about 8%) are below SDR=-5 dB.

The details of parameter range are summarized in Table 9.

Table 9: List of applied distortions with probabilities and parameter ranges.

| Augmentation | Prob. | Param. name | Range / Values | Notes |
|---|---|---|---|---|
| RIR convolution | 0.50 | - | - | direct-path delay compensated |
| Sample Noise | 1.00 | SNR (dB) | $[5, 20]$ | from DNS dataset |
| Colored Gaussian Noise | 1.00 | SNR (dB) | $[5, 20]$ | |
| | | exponent $\beta$ | $[0.75, 1.5]$ | |
| Band-limiting (BPF) | 0.20 | $f_1$ (Hz) | $[2000, 4000]$ | zero-phase |
| (occlusion FIR) | | $f_2$ (Hz) | $f_1 + [200, 500]$ | transition band upper edge |
| | | cut_gain | $(0.1, 0.3)$ | stopband gain, applied as $g^\beta$ |
| | | $\beta$ | $[0.25, 0.75]$ | (thus effective stopband $\approx [0.22, 0.55]$) |
| | | taps | odd in $[31, 61]$ | firwin2, $f_s$=16k ( $f_N$=8k) |
| Clipping | 0.20 | level (dB) | $[-10, 0]$ | hard clipping threshold |
| Crystalizer | 0.10 | intensity | $[1, 2]$ | spectral "sharpening" |
| Flanger | 0.05 | depth | $[1, 3]$ | short-delay comb filtering |
| Crusher (bit-depth) | 0.10 | bits | $[1, 5]$ | quantization/aliasing |
| Codec (any) | 0.25 | — | — | one of the following |
| MP3 | | bit rate (kbps) | $[16, 64]$ | variable bit-rate sampled uniformly |
| OGG | | encoder | vorbis, opus | random choice |
| Frequency Masking | 1.00 | $F_{bw}$ (freq. bins) | $[0, 5]$ | |
| | | # masks | $[0, 1]$ | |
| Time Masking | 1.00 | $T_{dur}$ (frames) | $[0, 5]$ | |
| | | # masks | $[0, 1]$ | |

# E  ABLATION DETAILS

## E.1  EFFECTS OF SCALED LOG-SPECTRAL LOSS

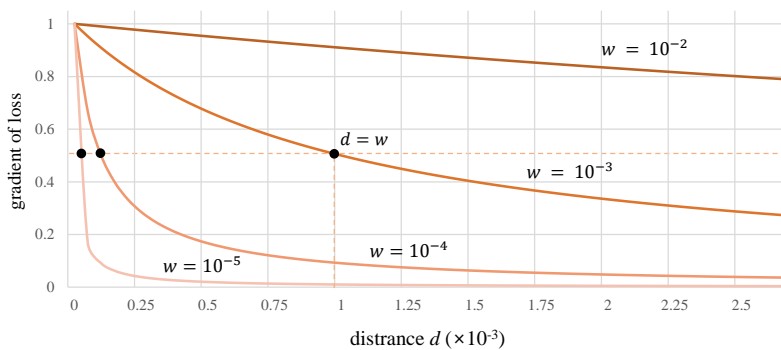

Figure 5: Gradient profiles of the proposed scaled log-spectral loss $\partial\ell/\partial d = w/(d + w)$ for different scale factors $w$. The curves show that the gradient is 1 near zero error and monotonically decreases as the distance $d = |y - s|$ grows. Smaller $w$ values make the loss more sensitive to fine spectral deviations, while larger $w$ values maintain stronger gradients over broader error ranges.

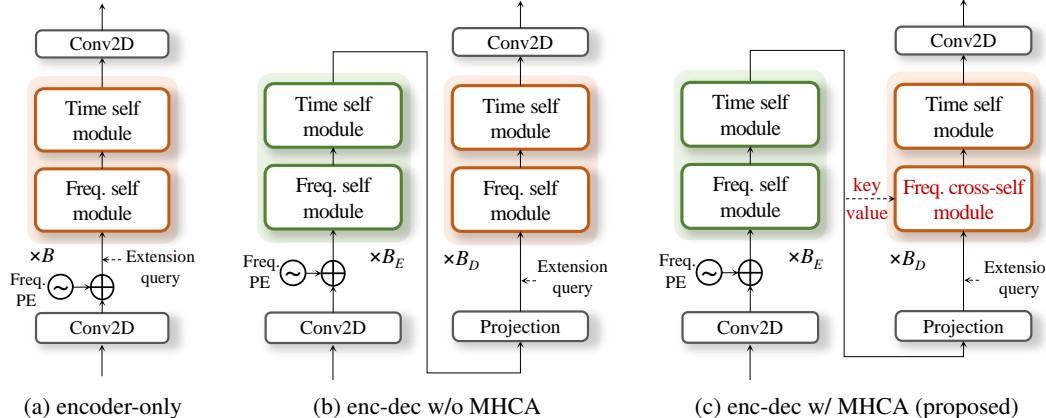

Figure 6: **Unit modules in TF-Encoder and TF-Decoder.** The (a) time module is based on MHSA with RoPE while (b) the frequency encoder module is based on MHSA with frequency projection layer. (c) The frequency decoder module utilize MHCA based on key/value from the encoder features

Figure 5 illustrates the gradient behavior of the proposed scaled log-spectral loss, defined as $\partial\ell/\partial d = w/(d+w)$ where $d = |y - s|$ denotes the spectral distance and $w$ is a scale factor. Unlike conventional $\ell_1$ or $\ell_2$ criteria, whose gradients are either constant regardless of error magnitude ($\ell_1$) or increase proportionally with larger errors ($\ell_2$), the proposed formulation yields gradients that are strongest near $d \approx 0$ and gradually diminish once $d$ exceeds $w$. This mechanism emphasizes regions where the spectrum is already well-aligned, thereby preserving fine details, while suppressing unstable updates from heavily corrupted regions. The figure shows that when $d = w$, the gradient magnitude stabilizes at $0.5$, providing a natural balance between emphasizing accurate components and de-emphasizing severely mismatched ones. As $w$ decreases, the loss becomes more sensitive to smaller deviations, further suggesting subtle spectral structures that are otherwise neglected in conventional losses.

### E.2 ILLUSTRATION OF ENCODER-DECODER STRUCTURE ABLATION

Figure 6 provides detailed comparisons of the three decoder designs considered in our ablation.

**(a) Decoder-only.** This variant directly inserts extension queries into the decoder without an encoder counterpart. The decoder therefore bears the full burden of modeling both the observed input band and the missing high-frequency bands, resulting in heavier computation and weaker inductive bias from the input.

**(b) Encoder-decoder without MHCA.** Here the encoder first analyzes the input bandwidth, and the decoder has the same internal structure as the encoder but receives projected extension queries. Although this design separates analysis and reconstruction, the decoder relies only on self-attention within the extended sequence, and does not explicitly exploit encoder features for reconstruction.

**(c) Encoder-decoder with MHCA (proposed).** In our final design, the decoder additionally uses a frequency cross-self module, where encoder outputs serve as key-value inputs for cross-attention while extension queries act as queries. This enables direct conditioning of high-frequency synthesis on encoder features, while the self-attention within the decoder refines spectral structure among extended bins. As a result, the encoder specializes in processing the observed input, and the lightweight decoder focuses on plausible high-frequency generation guided by encoder information.

These illustrations highlight how the proposed encoder-decoder with MHCA achieves a clear division of labor: the encoder concentrates on input-bandwidth analysis, and the decoder selectively extends spectral content with cross-conditioning, leading to better efficiency and stability compared to the other two designs.

## F EXTENSION QUERY UNDER IMBALANCED SAMPLING-RATE DISTRIBUTIONS

Table 10 analyzes whether extension-query tokens become undertrained when certain sampling rates appear too infrequently during training. For the input-rate ablation (top subtable), performance remains nearly identical to the balanced baseline as long as 8 kHz inputs constitute at least 10% of the data. The differences across {0.10, 0.25, 0.50} distributions are minimal in all target-rate settings, and even the best scores often occur at moderately imbalanced ratios. Noticeable degradation appears only in the extreme 1% case, where the model sees almost no examples of low-band inputs; this leads to modest but consistent drops, particularly for the largest gap ($8 \rightarrow 44.1$ kHz).

A similar pattern is observed for the output-rate ablation (bottom subtable). When high-frequency target rates (44.1 kHz or 48 kHz) have extremely low probability (1–5%), reconstruction quality decreases for those specific targets, as seen in elevated LSD/MCD values. However, once each output rate is represented with a reasonable frequency (around 10% or more), the performance aligns closely with the uniformly balanced case, and the differences across distributions remain small.

Overall, these results show that extension-query undertraining affects performance only under highly skewed sampling-rate distributions. TF-Restormer remains robust as long as each rate appears with moderate frequency, and balanced or mildly imbalanced settings exhibit negligible differences from the baseline.

Table 10: Ablation study on input/output sampling-rate distributions to evaluate the robustness of extension-query training. Grey rows denote the default distribution used for baseline models.

| $f_E$ (Hz) | | $8 \rightarrow 16$kHz | | | $8 \rightarrow 44.1$kHz | | | $16 \rightarrow 48$kHz | | |
|---|---|---|---|---|---|---|---|---|---|---|
| 8k | 16k | LSD$^\downarrow$ | MCD$^\downarrow$ | NISQA$^\uparrow$ | LSD$^\downarrow$ | MCD$^\downarrow$ | NISQA$^\uparrow$ | LSD$^\downarrow$ | MCD$^\downarrow$ | NISQA$^\uparrow$ |
| Input | | 3.36 | 11.38 | 1.91 | 3.64 | 11.47 | 1.91 | 3.48 | 11.37 | 1.73 |
| 0.01 | 0.99 | 1.48 | 5.83 | 4.01 | 1.46 | 7.12 | 4.41 | **1.15** | **2.82** | 4.55 |
| 0.10 | 0.90 | 1.21 | 2.82 | 4.48 | 1.24 | 3.20 | 4.46 | 1.17 | 2.85 | **4.56** |
| 0.25 | 0.75 | 1.16 | 2.78 | 4.49 | 1.18 | 3.08 | **4.52** | 1.18 | 2.86 | 4.54 |
| 0.50 | 0.50 | **1.14** | **2.74** | **4.50** | 1.17 | **3.05** | 4.52 | 1.20 | 2.89 | 4.53 |

(a) ablation of training $f_E$ distribution.

| $f_D$ (Hz) | | | | $8 \rightarrow 16$kHz | | | $8 \rightarrow 24$kHz | | | $8 \rightarrow 44.1$kHz | | | $16 \rightarrow 48$kHz | | |
|---|---|---|---|---|---|---|---|---|---|---|---|---|---|---|---|
| 16k | 24k | 44.1k | 48k | LSD$^\downarrow$ | MCD$^\downarrow$ | NISQA$^\uparrow$ | LSD$^\downarrow$ | MCD$^\downarrow$ | NISQA$^\uparrow$ | LSD$^\downarrow$ | MCD$^\downarrow$ | NISQA$^\uparrow$ | LSD$^\downarrow$ | MCD$^\downarrow$ | NISQA$^\uparrow$ |
| Input | | | | 3.36 | 11.38 | 1.91 | 3.49 | 11.60 | 1.91 | 3.64 | 11.47 | 1.91 | 3.48 | 11.37 | 1.73 |
| 0.49 | 0.49 | 0.01 | 0.01 | **1.15** | **2.73** | **4.51** | 1.17 | 3.05 | **4.52** | 1.36 | 4.01 | 4.49 | 1.51 | 6.83 | 4.01 |
| 0.45 | 0.45 | 0.05 | 0.05 | **1.15** | 2.74 | **4.51** | **1.16** | 3.05 | 4.51 | 1.20 | 3.16 | 4.49 | 1.27 | 3.00 | 4.46 |
| 0.40 | 0.40 | 0.10 | 0.10 | 1.16 | 2.76 | 4.50 | **1.16** | **3.04** | 4.51 | 1.21 | 3.15 | 4.49 | 1.23 | 2.94 | 4.49 |
| 0.25 | 0.25 | 0.25 | 0.25 | 1.16 | 2.78 | 4.49 | 1.18 | 3.08 | **4.52** | **1.18** | **3.08** | **4.52** | **1.18** | **2.86** | **4.54** |

(b) ablation of training $f_D$ distribution

## G EVALUATION OF DEDICATED MODEL ON THE URGENT CHALLENGE

In the main paper, we reported non-intrusive MOS results on the URGENT blind test set using the unified TF-Restormer model. Since the blind set does not provide clean references, only MOS-based evaluation is possible. For completeness, we additionally train a dedicated URGENT model and report objective fidelity metrics on the non-blind validation set in the challenge.

For the dedicated URGENT configuration, we follow the official training recipe. Because the benchmark resamples all inputs to the target sampling rate regardless of their original rate, we remove high-frequency bins with negligible power prior to processing, which reduces redundant computation under this matched-rate setting. Apart from this preprocessing, the model is trained under the same conditions required by the challenge.

It is important to note that the URGENT benchmark emphasizes deterministic, fidelity-oriented enhancement (Sun et al., 2025; Chao et al., 2025; Rong et al., 2025). Intrusive metrics are heavily weighted, and top-ranked systems typically rely on large, deterministic architectures optimized exclusively for matched-rate denoising. In contrast, models that based on generative approach to improve perceptual quality generally achieve lower intrusive scores despite producing more natural listening quality.

Under such conditions, a dedicated TF-Restormer variant trained with the URGENT recipe shows moderate improvement in signal fidelity compared to the unified model; however, its performance remains below that of highly specialized deterministic and multi-stage systems and perceptual quality become significantly low. This outcome is expected: the strengths of TF-Restormer—arbitrary input–output sampling-rate handling, frequency-extension mechanisms, perceptually oriented losses, and adversarial training—do not align with the evaluation objectives of URGENT. Consequently, while the unified model yields strong non-intrusive MOS on the blind test set, the dedicated URGENT-trained version does not fully reflect the core advantages of our architecture.

Table 11: Evaluation results on URGENT 2025 (non-blind test set)

| Model(Team) | Model Type | Rank | intrusive signal fidelity | | | | | Semantic fidelity | | | Non-intrusive quality | | |
|---|---|---|---|---|---|---|---|---|---|---|---|---|---|
| | | | PESQ$^\uparrow$ | ESTOI$^\uparrow$ | SDR$^\uparrow$ | MCD$^\downarrow$ | LSD$^\downarrow$ | sBERT$^\uparrow$ | SpkSim$^\uparrow$ | CAcc(%)$^\uparrow$ | UTMOS$^\uparrow$ | NISQA$^\uparrow$ | DNSMOS$^\uparrow$ |
| Input | - | - | 1.37 | 0.61 | 2.53 | 7.92 | 5.51 | 0.75 | 0.63 | 81.29 | 1.56 | 1.69 | 1.84 |
| baseline | D | 9 | 2.43 | 0.80 | 11.29 | 3.32 | 2.84 | 0.86 | 0.80 | 84.96 | 2.11 | 2.89 | 2.94 |
| Bobbsun | D | 1 | **2.95** | **0.86** | **14.33** | 3.01 | 2.83 | **0.91** | **0.85** | **88.92** | 2.09 | 3.22 | 2.88 |
| USEM(rc) | D | 2 | 2.79 | 0.85 | 13.11 | **2.93** | **2.94** | 0.90 | 0.84 | 88.05 | 2.30 | 3.21 | 3.01 |
| USEM-Flow(rc) | G | - | 1.54 | 0.59 | 4.49 | 6.10 | 3.91 | 0.76 | 0.66 | 69.07 | 1.79 | 2.82 | 2.50 |
| TS-URGENet(Xiaobin) | D+G | 3 | 2.74 | 0.84 | 13.06 | 3.30 | 3.08 | 0.89 | 0.84 | 87.94 | 2.16 | 3.24 | 2.92 |
| alindborg | D+G | 10 | 1.99 | 0.76 | 7.49 | 4.51 | 3.73 | 0.84 | 0.77 | 81.70 | 2.49 | 3.96 | **3.28** |
| wataru9871 | G | 13 | 1.36 | 0.56 | -13.88 | 11.25 | 7.98 | 0.82 | 0.51 | 79.70 | 2.53 | 3.74 | 3.10 |
| TF-Restormer (*original*) | D+G | - | 1.71 | 0.71 | 4.19 | 4.84 | 4.32 | 0.84 | 0.73 | 80.21 | **3.57** | **4.51** | 3.25 |
| TF-Restormer (*dedicated*) | D+G | - | 2.60 | 0.83 | 11.78 | 3.18 | 2.91 | 0.88 | 0.82 | 84.28 | 2.47 | 3.43 | 3.06 |

# H COMPARISON WITH CONVENTIONAL STREAMING ENHANCEMENT MODELS

We evaluate TF-Restormer-*streaming* against representative real-time denoising models on the VCTK+DEMAND test set, using PESQ, STOI, and the composite metrics CSIG, CBAK, and COVL commonly adopted in prior enhancement works. Unlike conventional streaming models, which are trained specifically for denoising under matched conditions, TF-Restormer-*streaming* inherits the full unified restoration and bandwidth-extension objective and is trained to handle reverberation, distortion, and bandwidth mismatch simultaneously. As a consequence, its model size and computational cost are substantially larger than lightweight denoisers such as NSNet2, DCCRN, or DeepFilterNet.

In terms of signal fidelity, specialized denoising models remain strong, with FRCRN and DeepFilterNet2 achieving the highest PESQ, CBAK, or STOI scores. Nevertheless, TF-Restormer-*streaming* attains competitive perceptual quality, achieving the highest CSIG score among all models and CBAK/COVL values close to the best discriminative systems. This is notable given that TF-Restormer-*streaming* is not optimized for denoising alone, but operates as a general-purpose restoration model that simultaneously handles reverberant, noisy, and bandwidth-limited inputs.

Overall, these results show that TF-Restormer-*streaming* is, to our knowledge, the first unified restoration and super-resolution model capable of streaming operation, while still providing signal fidelity comparable to denoising-oriented baselines. This demonstrates the feasibility of extending multi-rate, multi-distortion restoration models to real-time settings without sacrificing robustness.

Table 12: Comparison of TF-Restormer-*streaming* with existing real-time denoising models on the VCTK+DEMAND test set. We report standard enhancement metrics (PESQ, STOI, CSIG, CBAK, COVL) along with model size and MACs.

| Model | Size (M) | MACs (G/s) | PESQ | CSIG | CBAK | COVL | STOI |
|---|---|---|---|---|---|---|---|
| Noisy | - | - | 1.97 | 3.34 | 2.44 | 2.63 | 0.921 |
| NSNet2 (Braun et al., 2021) | 6.2 | 0.43 | 2.47 | 3.23 | 2.99 | 2.90 | 0.903 |
| DCCRN (Hu et al., 2020) | 3.7 | 14.36 | 2.54 | 3.74 | 3.13 | 2.75 | 0.938 |
| FullSubNet+ (Chen et al., 2022a) | 8.7 | 30.06 | 2.88 | 3.86 | 3.42 | 3.57 | 0.940 |
| FRCRN (Zhao et al., 2021) | 10.3 | 12.3 | **3.21** | 4.23 | **3.64** | **3.73** | - |
| DeepFilterNet (Schroter et al., 2022) | 1.8 | 0.11 | 2.81 | 4.14 | 3.31 | 3.46 | **0.942** |
| DeepFilterNet2 (Schröter et al., 2022) | 2.3 | 0.356 | 3.08 | 4.30 | 3.40 | 3.70 | 0.941 |
| TF-Restormer-*streaming* | 19.0 | 214.5 | 2.89 | **4.37** | 3.41 | 3.68 | 0.937 |

# I    EVALUATION ON DNS CHALLENGE DATASET

We further evaluate TF-Restormer on the 2020 DNS (Reddy et al., 2020) test sets to examine both signal fidelity and perceptual quality. Table 13 compares objective fidelity metrics against models specifically optimized for denoising with DNS training dataset. Since our unified model is trained to remove reverberation as well as noise, we report fidelity scores only on the "No Reverb" subset, where the clean references are aligned with our training objective. Under this setting, the unified TF-Restormer shows lower fidelity than DNS-targeted systems such as MFNet, USES, and TF-Locoformer. This gap is expected, as the unified model (i) is trained solely on VCTK, (ii) actively removes reverberation and other distortions, and (iii) incorporates perceptual objectives that may deviate from strict waveform fidelity. When trained in a DNS-specific manner without adversarial objectives, however, the dedicated TF-Restormer variant matches or surpasses prior systems, achieving competitive PESQ, STOI, and SI-SDR scores.

Table 14 presents perceptual DNSMOS scores on both "With Reverb" and "No Reverb" subsets. Here, discriminative models tend to preserve the input structure and thus achieve relatively conservative perceptual gains, as seen with Conv-TasNet and FRCRN. Generative approaches such as SELM, GenSE, MaskSR, and UniSE, which prioritize perceptual naturalness, obtain noticeably higher OVRL scores. TF-Restormer shows perceptual quality on par with these generative systems across all subsets, despite not being trained exclusively for perceptual enhancement. In both reverberant and non-reverberant conditions, it achieves strong SIG and BAK scores and matches the best OVRL scores among recent models, demonstrating that the proposed architecture can deliver high perceptual quality while maintaining reasonable signal fidelity.

Table 13: Comparison of TF-Restormer with previous models on 2020 DNS testsets in terms of signal fidelity. "No Reverb" subset is only compared as the proposed TF-Restormer is trained to remove reverberation.

| System | PESQ | STOI(%) | SI-SDR(dB) |
|---|---|---|---|
| Noisy | 1.58 | 91.5 | 9.1 |
| FullSubNet (Hao et al., 2021) | 2.78 | 96.1 | 17.3 |
| CTSNet (Li et al., 2021) | 2.94 | 96.2 | 16.7 |
| TaylorSENet (Li et al., 2022) | 3.22 | 97.4 | 19.2 |
| FRCRN (Zhao et al., 2021) | 3.23 | 97.7 | 19.8 |
| MFNet (Liu et al., 2023) | 3.43 | 97.9 | 20.3 |
| USES (Zhang et al., 2023) | 3.46 | 98.1 | 21.2 |
| TF-Locoformer (Saijo et al., 2024) | 3.72 | 98.8 | **23.3** |
| TF-Restormer | 2.83 | 96.4 | 16.1 |

Table 14: Comparison of TF-Restormer with previous models on 2020 DNS testsets in terms of perceptual quality (DNS scores). "With Reverb" subset contains reverberation while "No Reverb" subset only involves noise. "D" and "G" denote discriminative and generative methods, respectively.

| Model | Type | With Reverb | | | No Reverb | | |
|---|---|---|---|---|---|---|---|
| | | SIG | BAK | OVRL | SIG | BAK | OVRL |
| Noisy | - | 1.76 | 1.50 | 1.39 | 3.39 | 2.62 | 2.48 |
| Conv-TasNet (Luo & Mesgarani, 2019) | D | 2.42 | 2.71 | 2.01 | 3.09 | 3.34 | 3.00 |
| FRCRN (Zhao et al., 2021) | D | 2.93 | 2.92 | 2.28 | 3.58 | 4.13 | 3.34 |
| SELM (Wang et al., 2024) | G | 3.16 | 3.58 | 2.70 | 3.51 | 4.10 | 3.26 |
| MaskSR (Li et al., 2024) | G | 3.53 | 4.07 | 3.25 | 3.59 | 4.12 | 3.34 |
| AnyEnhance (Zhang et al., 2025) | G | 3.50 | 4.04 | 3.20 | 3.64 | **4.18** | 3.42 |
| GenSE Yao et al. (2025) | G | 3.49 | 3.73 | 3.19 | 3.65 | **4.18** | **3.43** |
| LLaSE-G1 (Kang et al., 2025) | G | 3.59 | 4.10 | 3.33 | **3.66** | 4.17 | 3.42 |
| UniSE(Yan et al., 2025) | G | **3.67** | 4.10 | **3.40** | **3.67** | 4.14 | **3.43** |
| TF-Restormer | D+G | 3.60 | **4.12** | 3.35 | 3.65 | **4.18** | **3.43** |

## J    Limitations and Future Works

Although TF-Restormer demonstrates balanced improvements in both signal fidelity and perceptual quality, several limitations remain.

First, when the input speech is extremely degraded, the uncertainty in the observed spectrum can lead to content or speaker artefacts. This reflects a fundamental trade-off in speech restoration: adversarial training promotes perceptually natural reconstructions by hallucinating plausible high-frequency details, whereas purely supervised objectives remain closer to the reference but often oversmooth, reducing naturalness.

Second, the model may exhibit mild rate-distribution sensitivity. As discussed in Appendix F, severely imbalanced exposure to certain sampling rates (e.g., $<1\%$) can lead to undertrained extension-query regions, particularly at the highest frequencies. Although moderate coverage ($\approx 10\%$) is sufficient in practice, this highlights a remaining limitation of our arbitrary-rate formulation.

Thrid, because the model is trained on heavily distorted scenarios using limited single-language clean corpora, it may inherit language-specific biases, particularly due to the perceptual loss's dependency on a pretrained speech model (e.g., WavLM). This could affect cross-lingual generalization to underrepresented phonetic patterns.

Finally, while TF-Restormer addresses universal restoration, it does not handle multi-speaker conditions such as speech separation or speaker extraction. The current formulation assumes a single target speaker and does not incorporate mechanisms for resolving overlapping speech.

Future work includes reducing hallucination artefacts under extreme degradations, improving robustness under highly imbalanced sampling-rate distributions, extending the training pipeline to multilingual and more diverse corpora, and integrating the framework with multi-speaker modeling. Given that the TF dual-path architecture was originally proposed for separation, the proposed model has promising potential for extension to overlapped multi-speaker restoration.

