# OpenReview forum: "TF-Restormer: Complex Spectral Prediction for Speech Restoration"
_ICLR.cc/2026/Conference — Submitted to ICLR 2026_

### Official Review · Reviewer_HvyR · 2025-10-22

**Soundness:** 2
**Presentation:** 2
**Contribution:** 2
**Rating:** 2
**Confidence:** 5

**Summary:**

This paper proposes a universal speech enhancement model called TF-Restormer, which reconstructs missing high-frequency bands through a lightweight decoder with frequency extension queries. It enables efficient and universal restoration across arbitrary input–output rates without redundant resampling and supports streaming with a causal time module. The results shown in the experimental tables and demo audio are good. However, several important concerns regarding the experimental setup and comparisons should be addressed.

**Strengths:**

1)	The paper tackles an important and often overlooked issue in speech restoration: achieving a good balance between signal fidelity and perceptual quality. The authors explicitly consider both aspects during training.
2)	The concept of frequency extension queries is interesting and reasonable, though some implementation details are missing.
3)	The proposed frequency cross-self module and the use of inductive bias for structural consistency are also intriguing (but I feel the presentation is not very clear).

**Weaknesses:**

1)	The most serious concern is that the baselines reported in Tables 1 to 3 appear to be trained on different datasets and degradation conditions, making it unclear whether the performance gains come from the model architecture or the training data. I suggest training and evaluating TF-Restormer on a standardized benchmark, such as the URGENT Challenge (https://urgent-challenge.github.io/urgent2025/), for a fair comparison.
2)	The explanation of the shared projection layer for structural bias is unclear. While it may reduce model size, it’s not obvious for me how it improves performance. Although the ablation in Table 4(c) supports its effectiveness, it would be informative to show results when frequency projection is applied but **not** shared across all modules and key–value mappings.
3)	Although many previous works have already used log1p to compress the dynamic range so that the model is not dominated by high-energy regions, the rationale behind the proposed scaled log-spectral loss does not seem entirely convincing. It produces “gradients that are strongest near d≈0 and gradually diminish”, which somewhat appears to contradict the claim that it “emphasizes regions where the spectrum is already well-aligned.” In fact, it may have the opposite effect. How do you ensure that the large gradients do not cause the loss in “well-aligned regions” to oscillate around the optimal point?
4)	In Table 3 (super-resolution evaluation), UTMOS scores should not be reported since the metric always resamples audio to 16 kHz. It would be better to report NISQA or other suitable metrics instead.

5)	Several references related to the URGENT Challenge and the universal speech enhancement models submitted to it are missing:
Ex:


  [1] “Interspeech 2025 urgent speech enhancement challenge.” In Proc. Interspeech, 2025.


  [2] “Scaling beyond denoising: Submitted system and findings in urgent challenge 2025”, In Proc. Interspeech, 2025.


  [3] “Universal speech enhancement with regression and generative mamba”, In Proc. Interspeech, 2025.

**Questions:**

1)	For the ablation study, did you use fixed training steps (i.e.,100,000 for pretraining and 10,000 for adversarial training)? For a fair comparison, I would suggest determining the training steps based on a validation set, as different architectures may have different optimal training steps.
2)	For the learned frequency extension queries, do you learn only a single vector (as you mentioned shared across frames)? Under different input/target sampling rate settings, do you use different vectors (e.g., one for 16→32 kHz and another for 8→48 kHz)?
3)	Why MHCA only “operates when extension query is padded, otherwise, bypassed.”?

---

> ### Author Response · Authors · 2025-11-25
>
> # **Comments on Weaknesses**
>
> ---
>
> ### **(1) Concerns on experimental fairness**
>
> We thank the reviewer for raising this important point. We agree that it is difficult to fully separate architectural gains from potential advantages introduced by the training pipeline.
>
> To address this, we first evaluated the **same unified TF-Restormer** from Tables 1–3—trained once on VCTK clean sources—on three external test sets in Table 4 (VoxCeleb, LibriTTS, URGENT blind). This shows that the unified model generalizes well to **real-recorded speech** and **multi-corpus mixtures** despite being trained on a single corpus.
>
> Second, and more importantly for fairness, we trained **dedicated TF-Restormer variants** using distortion pipelines that exactly match those of the baseline methods. These results are added to **Tables 2 and 3**. As expected, the dedicated variants outperform the unified model on their corresponding tasks, confirming that the unified model operates under **strictly more challenging conditions**, and that its performance gains do not come from training-data advantages.
>
> Third, following the reviewer’s suggestion, we also trained a model fully aligned with the URGENT Challenge (multi-corpus training and matched-rate enhancement), and the results are reported in Appendix G. As stated in the general response, URGENT evaluates only the matched-rate, deterministic enhancement aspect, whereas TF-Restormer is designed for arbitrary-rate reconstructive restoration. Accordingly, our URGENT-trained variant achieves competitive non-intrusive perceptual quality within the scope of that benchmark.
>
> We hope these expanded evaluations resolve the fairness concerns by demonstrating performance under (1) unified multi-task conditions, (2) task-matched dedicated pipelines, and (3) standardized URGENT settings.
>
> ### **(2) Effectiveness of the frequency-projection layer**
>
> Thank you for the suggestion. We added the **non-shared (separate)** projection variant to Table 4(c). While both shared and separate projections outperform the no-projection baseline, the separate version shows **less stable behavior**—especially in SSR—and roughly doubles the parameter count (**30.1 M → 63.6 M**). Given the comparable performance but better stability and parameter efficiency, the **shared projection layer** is the more practical design.
>
> ### **(3) Effectiveness of the proposed loss**
>
> We expanded the loss ablations as requested (Table 5). These experiments clarify how spectral terms interact with the perceptual loss and why adaptive scaling is needed.
>
> Importantly, the **gradient of the scaled log-spectral loss is always bounded by 1**, identical to ℓ₁, so it does *not* cause instability near d≈0. In practice, the perceptual loss acts as a **complementary stabilizing objective**, preventing optimization drift and improving convergence behavior.
>
> Fixed-scale log-spectral variants behave either **too similarly to ℓ₁** (when the scale is large) or **produce vanishing gradients** (when the scale is small).
>
> ### **(4) UTMOS in Table 3**
>
> Thank you for the observation. As the reviewer noted, UTMOS is not appropriate in this setting. We therefore replaced it with **NISQA**, which operates directly at the target bandwidths and is better suited for our arbitrary-rate evaluation.
>
> ### **(5) References related to the URGENT Challenge**
>
> We have added all relevant URGENT references, along with the corresponding URGENT-style training and evaluation results in the Appendix.
>
> # **Comments on Questions**
>
> ---
>
> ### **(1) Training steps and model selection**
>
> All ablation variants were trained with a fixed number of steps that we verified to be sufficient for convergence. In practice, the validation loss for all architectures plateaued around 60k–70k steps, and we did not observe signs of overfitting beyond this point. We have clarified this procedure in the revised text.
>
> ### **(2) Whether extension queries are shared or rate-specific**
>
> We use a **single unified extension-query vector**, where each frequency bin has its own learnable entry.
> For different input/target sampling-rate settings, the decoder simply **selects the corresponding slice** of this vector to match the desired output bandwidth, enabling a continuous frequency representation without separate query sets.
> To improve clarity, we have also refined the manuscript description to more explicitly show how the unified query vector is appended along the frequency axis and interacts.
>
> ### **(3) Why MHCA is bypassed when extension queries are absent**
>
> MHCA is designed to operate **only on extension-query**, using them as queries and the encoder features as keys/values. When no extension queries are present—i.e., when the target bandwidth matches the input—there are simply **no query tokens for MHCA to process**. In this case, the module is skipped by design, and the decoder proceeds using MHSA on the observed-band features. We clarified this conditional behavior in the revision.

---

### Official Review · Reviewer_NvNb · 2025-10-29

**Soundness:** 3
**Presentation:** 4
**Contribution:** 3
**Rating:** 8
**Confidence:** 4

**Summary:**

The paper presents an encoder-decoder speech enhancement method that is capable of working with input audio with different sampling rates. The presented model is based on the Transformer architecture and consists of a heavy encoder and a light decoder. The training comprises two stages: the first stage involves the perceptual loss and the newly proposed log-spectral loss, which, as the authors claim, accounts for a better balance between well-predicted and poorly-predicted regions during training; the second stage uses SFI-STFT discriminators that enable adversarial training for varying frequencies. The model is evaluated on several key speech enhancement benchmarks.

**Strengths:**

There are several noteworthy strengths of the paper:

1. The presentation of the newly developed method is well-structured and clear, and the paper is well written. All method components are well-described, including detailed explanations of the model architecture, training procedure, and training losses. The provided codebase is very helpful in understanding the internal structure of the model.
2. The capability of the model to work on various sampling rates is a valuable contribution to the field.
3. The paper provides a thorough ablation study that justifies most of the design choices made in the paper.

**Weaknesses:**

However, there are several weak points:

1. The paper could benefit from additional benchmarking on several datasets, including VoxCeleb [1], LibriTTS [2], and DNS [3]. Moreover, other recent methods, such as GenHancer [5] and MIIPHER [4] could also be considered.
2. The padding in the decoder, which allows the model to be trained on recordings with various sampling rates, crucially relies on the model being trained on samples with both the smallest and the largest sampling rate. Otherwise, the pad tokens that correspond to the lowest or highest frequencies might be undertrained, resulting in worsened performance. This is a potential vulnerability of the model, and, therefore, should be flagged and investigated.

**Questions:**

1. It would be interesting to conduct an ablation study to see how the undertrained padding tokens can impact performance.
2. It would also be interesting to see how the use of perceptual loss impacts the quality of the model. I would welcome an additional ablation study on the use of perceptual loss in both pre-training and adversarial fine-tuning.
3. Since the authors claim that their method can be easily turned into a streaming algorithm, it would be interesting to compare the model to other streaming baselines, both in terms of quality and inference speed.

#### **References:**

[1] Nagrani et al., "VoxCeleb: a large-scale speaker identification dataset"

[2] Zen et al., "LibriTTS: A corpus derived from LibriSpeech for text-to-speech"

[3] Dubey et al., "ICASSP 2023 Deep Noise Suppression Challenge"

[4] Koizumi et al., "MIIPHER: a robust speech restoration model integrating self-supervised speech and text representations"

[5] Yang et al., "Genhancer: high-fidelity speech enhancement via generative modeling on discrete codec tokens"

---

> ### Author Response · Authors · 2025-11-25
>
> ## **Comments on Weaknesses**
>
> ---
>
> ### **(1) Additional benchmarking datasets and models**
>
> Thank you for the suggestion. We expanded our evaluation accordingly in Table 4. The **VoxCeleb** results, previously located in the Appendix, have been moved into the main paper since they provide meaningful evidence of real-recorded robustness. In Table 4, we additionally include evaluations on **LibriTTS** and the **URGENT blind test set**. We also report results on the **DNS 2020 dataset** in the Appendix I, covering both signal-fidelity and perceptual metrics (Tables 13 and 14).
>
> Regarding **MIIPHER**, the model implementation is not publicly available, but its *LibriTTS-R* dataset is. Therefore, we include comparisons on LibriTTS-R under the same data conditions. For **GenHancer**, neither a public implementation nor overlapping datasets are available, but we incorporated several recent strong baselines in Tables 4, 11, 13, and 14 to ensure broad and up-to-date comparisons.
>
> ### **(2) Potential undertraining of extension-query padding tokens**
>
> We appreciate the reviewer highlighting this subtle issue. In our main setup, all data originate from **48 kHz recordings**, and input/output sampling rates are **sampled uniformly**. Under this balanced configuration, we did not observe performance differences across output rates, indicating that extension-query tokens were sufficiently trained.
>
> That said, we agree with the reviewer that **imbalanced multi-rate distributions** can lead to undertrained tokens at extreme frequency ranges. We now explicitly discuss this potential vulnerability in the paper. Furthermore, we conducted additional experiments that vary the input/output sampling-rate distributions to directly test this behavior; the results (confirming the reviewer’s intuition) are included in the Appendix F.
>
> ## **Comments on Questions**
>
> ---
>
> ### **(1) Impact of undertrained extension-query tokens**
>
> We conducted controlled ablations varying both input-rate and output-rate sampling distributions (Table 10).
>
> - **Input-rate imbalance:**
>
>     When 8 kHz inputs appear only **1%**, the corresponding low-band extension queries become undertrained, producing modest degradation for the largest gap (8→44.1 kHz).
>
>     At **≥10%**, performance matches the balanced baseline across all target rates.
>
> - **Output-rate imbalance:**
>
>     Degradation appears only in the extreme **{49,49,1,1}** setting for {16k, 24k, 44.1k, 48k}, where the highest-rate extensions are rarely seen.
>
>     Even **{40,40,10,10}** is sufficient to maintain stable reconstruction.
>
>
> These results show that TF-Restormer is **robust** to sampling-rate imbalance: only very extreme (≈1%) cases cause noticeable drops, while **~10% exposure is already sufficient** to avoid undertrained extension queries.
>
> ### **(2) Impact of perceptual loss during pretraining and fine-tuning**
>
> As further detailed in the general response, we performed extensive ablations on the loss including ablations of perceptual loss in Table 5. Removing it consistently degraded perceptual quality, while combining it with the scaled log-spectral loss provided the best balance between fidelity and perceptual quality.
>
> ### **(3) Comparison with streaming baselines**
>
> To our knowledge, publicly available *streaming* systems for **general speech restoration**—handling denoising, dereverberation, bandwidth extension, and other distortions simultaneously—are extremely limited. Most existing streaming baselines are **denoising-only** models.
>
> Thus, in Appendix H, we compare TF-Restormer-streaming with representative denoising-oriented systems (NSNet2, DCCRN, FRCRN, DeepFilterNet). Since TF-Restormer-streaming inherits the **full restoration and bandwidth-extension objective**, its model size and computational cost are naturally larger, and direct comparisons in inference speed would not be meaningful. Instead, we report **model size and MACs**, which reflect the computational characteristics of the streaming variant. Despite this mismatch in design focus, TF-Restormer-streaming achieves **competitive perceptual quality** (e.g., the highest CSIG score in Table 12), demonstrating that the proposed architecture can be extended to streaming without sacrificing robustness.

---

> ### Comment · Reviewer_NvNb · 2025-11-26
>
> I would like to thank the authors for their responses. I do not have any further questions and am planning to maintain my score.

---

### Official Review · Reviewer_7rbh · 2025-11-01

**Soundness:** 2
**Presentation:** 3
**Contribution:** 2
**Rating:** 2
**Confidence:** 4

**Summary:**

This paper introduces a transformer-based architecture, TF-RESTORMER, for tackling the general speech restoration task. Inherited from the commonly adopted sampling-frequency independent strategy, this paper further extends to supporting arbitrary input-output sampling rates with a well-designed asymmetric TF dual path structure and proposes a projection-based spectral inductive bias. To further enhance speech quality, this paper proposes a scaled log-spectral loss. Extensive experiments have been conducted on general speech restoration, denoising, and super-resolution tasks to verify the effectiveness of TF-RESTORMER.

**Strengths:**

1. This paper further extends the commonly adopted sampling-frequency independent strategy with supporting arbitrary input-output sampling rates. Additionally, the proposed projection-based spectral inductive bias appears to be effective based on the experiments.

2. This paper proposes a scaled log-spectral loss for spectral supervision, which seems to be beneficial for improving the performance of speech restoration tasks and for the development of loss designs within the community.

**Weaknesses:**

1. This paper only uses clean source from the VCTK dataset, which is relatively small, comprising merely 44 hours of recordings. For the general speech restoration task, it is more common to employ a large-scale corpus augmented with diverse degradations. Additionally, the distortions added for mixing up the degraded signal are moderate. For example, while this paper limits the signal-to-noise ratio (SNR) to 5-20 dB, comparable studies evaluate severe noise occasions with SNR down to -5 dB or even -10 dB. The results obtained from such a small dataset and trained with moderate distortions, are not sufficiently convincing. And some testsets that are more widely adopted in the general speech restoration task, such as the testset from the URGENT Challenge, should be considered. Additionally, given that the TF-Restormer can simultaneously perform multiple tasks, including denoising and super-resolution, it is recommended to evaluate the model on different test sets using a single trained model.

2. This paper proposes an SFI-STFT Discriminator, yet it provides insufficient experimental results to prove its effectiveness. It's unclear whether using a shared SFI-STFT Discriminator for all the sampling rate settings is more effective than using a set of independent discriminators, each targeting a specific sampling rate. Without this ablation, the claimed superiority of the shared design in the discriminator remains unsubstantiated.

3. Although the paper introduces a scaled log-spectral loss, its effectiveness is not adequately demonstrated. The ablation study should be extended to include experiments in which the scale factor is held constant, thereby verifying the contribution of the scaling mechanism.

**Questions:**

1. The effectiveness of the shared SFI-STFT Discriminator remains unclear to me. I am skeptical that a shared SFI-STFT Discriminator can outperform a set of independent discriminators, each targeting a specific sampling rate, given that the latter actually involves the same training costs as the proposed SFI-STFT Discriminator but with specialized capacity for supervising the performance in each sampling rate. Therefore, I suggest conducting auxiliary experiments to compare these two approaches.

2. Does the author conduct auxiliary experiments to further investigate the proposed scaled log-spectral loss? For instance, fixing the value of the scale factor.

3. The formatting of the references is inconsistent and should be standardized.

---

> ### Author Response · Authors · 2025-11-25
>
> ## **Comments on Weaknesses**
>
> ### **(1) Evaluation setup, distortion severity, and URGENT Challenge**
>
> ---
>
> We sincerely thank the reviewer for the valuable comments. We agree with the concern that relying solely on VCTK clean sources for training and evaluation can be limiting for general speech restoration.
>
> Nevertheless, our first goal was to demonstrate that **a single unified TF-Restormer model** can operate robustly across heterogeneous sampling-rate and distortion conditions, and Tables 1–3 are structured to highlight this unified-model behavior.
>
> We also note that the GSR+SSR evaluation in Table 3 is **not moderate**. Although the additive noise components individually span SNR ∈ [5, 20] dB, they are applied **simultaneously** (sampled noise + Gaussian noise), and the unified pipeline additionally applies **reverberation**, **clipping**, **band-pass filtering**, **masking**, **codec artifacts**, and **random resampling**. As a result, the **effective distortion severity is much higher**: over the 2,937 evaluation samples, the input averages SDR = **2.11 dB**, **234 utterances (about  8%) below SDR=−5 dB**, and some cases exceed SDR=**−10 dB**. These statistics are comparable to the difficulty level of the URGENT benchmark.
>
> We fully agree with the reviewer that broader evaluation on diverse test sets is important. Following this suggestion, we added **Table 4**, which reports results from the **same single unified model on three additional test sets (VoxCeleb, LibriTTS, URGENT).** Across all three benchmarks, the unified model demonstrates competitive or superior performance to recent restoration systems, further supporting its generalization ability. We also include evaluations on DNS and experiments related to the URGENT Challenge in the Appendix to provide broader and more comprehensive evidence of the model’s robustness.
>
> ### **(2) Concerns about the shared SFI-STFT discriminator**
>
> We appreciate the reviewer’s suggestion. We previously explored **rate-specific discriminators**, but empirically they produced **less stable training** and **lower overall performance**. Because SFI-STFT provides a **rate-normalized TF representation**, the **physical meaning of TF patterns is consistent across sampling rates**, and CNN-based discriminators naturally assume **location-invariant semantics**. Splitting them across sampling rates therefore fragments the adversarial signal: each discriminator sees only a subset of TF conditions and gives less coherent feedback to a single unified generator.
>
> For completeness, we added an ablation comparing **shared** vs **non-shared** discriminators. The shared version provides the best balance of reconstruction quality, cross-rate consistency, and implementation simplicity.
>
> ### **(3) On the scaled log-spectral loss**
>
> As expanded in the general response, we included a range of **fixed scale-factor** ablations. Large constants behave similarly to L1 due to the typical spectral error scale (1e-2 to 1e-3), while very small constants suppress gradients prematurely. The **adaptive scaling** yields most stable and consistent improvements across frequency regions.
>
> ## **Comments on Questions**
>
> ---
>
> ### **(1) Effectiveness of the SFI-STFT discriminator**
>
> Addressed in Weakness (2). Table 6(a) includes direct comparisons showing that the shared discriminator provides more stable gradients, better cross-rate coherence, and comparable training cost.
>
> ### **(2) Additional experiments for the scaled log-spectral loss**
>
> Yes. All fixed-scale variants and comparisons requested by the reviewer are included in the Table 5 and discussed in the General Response.
>
> ### **(3) Reference formatting**
>
> We appreciate the reviewer’s attention. All references have been standardized according to the ICLR guidelines.

---

### Official Review · Reviewer_5YCi · 2025-11-03

**Soundness:** 2
**Presentation:** 2
**Contribution:** 2
**Rating:** 2
**Confidence:** 4

**Summary:**

This paper proposes TF-Restormer, an encoder-decoder architecture designed for universal speech restoration. The key innovation is its ability to handle arbitrary input-output sampling rates without external resampling. The model uses a Time-Frequency (TF) dual-path encoder to analyze the input bandwidth and a lightweight decoder that uses extension queries to reconstruct missing high-frequency bands. It also introduces a streaming variant using causal modules (Mamba blocks) and a scaled log-spectral loss to stabilize training under severe degradations.

**Strengths:**

- Practicality & Flexibility: The ability to handle various input/output rates in a single model without external resampling is a significant practical advantage over models that require fixed rates.
- Performance Claims: It claims to achieve a better balance of signal fidelity (PESQ, SDR) and perceptual quality (MOS metrics) compared to pure vocoder or diffusion baselines

**Weaknesses:**

- **Novelty**: While the combination is novel, the core components (encoder-decoder model, TF-dual path, conformer/mamba blocks, adversarial training) are established.

- **Fairness**: In the super-resolution task, standard dedicated models were compared. It is not clear if these baselines were also allowed to use similar advanced training data augmentations to ensure a fair comparison.

- **Complexity vs. Performance**: The ablation study (Table 4b) shows the encoder-decoder structure is better than a "decoder-only" approach. It is not clear if the added complexity of the asymmetric design is fully justified by the performance gains across all tasks.

**Questions:**

- **Novelty**: To what extent do you consider the TF-Restormer a novel architectural contribution versus a clever application of existing TF dual-path and seq2seq principles to the SFI problem? specifically, is the 'extension query' mechanism a significant departure from standard decoder practices?

- **Generalizability**: "The paper focuses heavily on speech-specific properties (e.g., frequency structures). In your view, are the core contributions—like the SFI formulation or Scaled Log-Spectral Loss—generic enough to benefit other fields dealing with time-series or spectral data?

- **Reproducibility**: Given the complexity of the training data simulation pipeline, are there any nuanced implementation details not currently covered in Appendix B that would be critical for exact reproduction of the reported baselines and results?

---

> ### Author Response · Authors · 2025-11-25
>
> ### Comments on Weaknesses
>
> ---
>
> ### **(1) Architectural novelty and design motivation**
>
> We appreciate the reviewer’s comments. As described in the general response, the novelty of TF-Restormer does not lie in isolated components, but in **restructuring the restoration pipeline to explicitly handle mismatched input–output bandwidths**. Prior systems typically assume fully padded spectra, whereas our model instead **decouples input-band analysis from output-band reconstruction**. Although the decoder uses cross-attention, it does not operate as a seq2seq decoder. It does not autoregressively generate a sequence; rather, following an MAE-style masked reconstruction philosophy, the model adopts an asymmetric encoder–decoder design in which the encoder analyzes only the observed spectrum and the decoder reconstructs the missing regions.
>
> The extension queries explicitly represent the *unobserved high-frequency region*, and their number is determined by the **user-specified target bandwidth**, not by a decoding schedule. Cross–self attention is used solely to condition these queries on the observed-band features, not to perform token-level generation.
>
> Thus, the contribution lies in an **architectural reframing** of the SFI setting. Taken together, these design choices represent a coherent shift from prior input–output–matched SFI or fixed-rate systems, enabling arbitrary-rate restoration within a single unified model.
>
> ### **(2) Clarification on SSR setup and fairness**
>
> As summarized in the general response, our evaluation uses **a single unified model** across all tasks, meaning TF-Restormer is *not* tuned for SSR specifically and is inherently disadvantaged against dedicated SSR baselines. Regarding the reviewer’s concerns, our augmentation strategy is designed for **general restoration**, and many elements (e.g., bandwidth perturbation, codec simulation) do not benefit SSR-specific models—and can in fact be detrimental when the target bandwidth is fixed.
>
> To address this concern directly, we additionally included **dedicated TF-Restormer variants** trained under SSR distortion pipelines, following the protocols used by baseline models (Table 3). These results show that the unified model is conservative, while the task-matched variants perform better, ensuring fairness while preserving the intent of demonstrating unified arbitrary-rate restoration.
>
> ### **(3) On the decoder-only baseline**
>
> To clarify, the decoder-only baseline is not simply the proposed model with the encoder removed. For a fair comparison, we constructed a decoder-only architecture whose computational cost matches that of the full model. While the proposed encoder–decoder uses B_E = 6 encoder blocks and B_D=3 decoder blocks, the decoder-only model uses B_D=9 blocks to equalize average MACs across settings.
>
> To make this comparison explicit, Table 4(b) reports both the parameter size and the MACs for each configuration. Because the decoder-only architecture employs smaller channel dimensions, its parameter count is smaller, but depending on the condition its MACs can be comparable or even higher than those of the proposed model.
>
> To further isolate the effect of architectural design rather than capacity, we additionally evaluated a **size-matched encoder–decoder variant**, where the encoder channel is reduced to match the parameter count of the decoder-only baseline. Even in this parameter-matched setting—where the proposed model uses **significantly fewer MACs**—the encoder–decoder architecture consistently outperforms the decoder-only design.
>
> These results collectively confirm that the observed performance improvements do not arise from model size or computational budget, but from the effectiveness of the **asymmetric encoder–decoder structure**, which provides a more suitable decomposition for sub-band analysis and high-frequency reconstruction in SSR tasks.
>
> ## Comments on Questions
>
> ---
>
> ### **(1) Extension query and novelty**
>
> The extension queries are not decoder tokens in the seq2seq sense. They parameterize the *unobserved frequency region* of the target bandwidth, with their length directly determined by the user-specified output sampling rate. Through cross–self attention, they enable **masked spectral completion** conditioned on the observed-band features—fundamentally different from padding-based or seq2seq decoding mechanisms.
>
> ### **(2) Generalizability**
>
> Both the SFI formulation and the scaled log-spectral loss are modality-agnostic. The representation naturally extends to other TF-domain signals (e.g., biomedical, vibration, radar), and the proposed loss stabilizes spectral regression across domains.
>
> ### **(3) Reproducibility**
>
> All augmentation parameters, ranges, sampling procedures, and codec settings are detailed in Appendix B. The anonymous code repository contains complete scripts for data simulation, training, and evaluation. Pretrained weights will also be released to ensure exact reproducibility.

---

> ### Author Response · Authors · 2025-11-28
> **A Clarification for the terminology "decoder-only" baseline**
>
> Upon further reflection on the reviewer’s feedback, we recognized that the term “decoder-only baseline” in the ablation study could naturally invite ambiguity, as the word "decoder-only" is often associated with seq2seq or autoregressive architectures.
> To avoid any unintended interpretation, we have revised the manuscript to use the term “encoder-only baseline.” We also refer readers to the structural diagrams in the Appendix for a clearer depiction of how each baseline is constructed.
>
> This terminology more precisely reflects the intended role of this baseline—namely, a model that performs full-band reconstruction without the proposed bandwidth-aware encoder.
> We hope this refinement helps ensure that the architectural comparison is interpreted as originally intended and offers a clearer distinction between the encoder’s bandwidth-analysis role and the decoder’s masked spectral completion through extension queries.

---

### Author Response · Authors · 2025-11-25
**General Response to All Reviewers**

We sincerely thank all reviewers for their thoughtful and constructive feedback. The comments consistently raised important questions regarding our evaluation protocol, fairness of comparisons, architectural choices, and the behavior of proposed losses and modules. We have revised the manuscript accordingly and expanded our experiments to address these points comprehensively.

---

## **(1) Clarifying the Main Contribution and Evaluation Protocol**

A key clarification is that TF-Restormer is designed as a **single unified model** capable of **arbitrary-input / arbitrary-output sampling-rate restoration**, rather than a collection of per-task systems. Its asymmetric encoder–decoder design separates **input-bandwidth analysis** (encoder) from **high-frequency reconstruction** via **extension queries** (decoder), avoiding the fixed-rate assumptions common in prior works.

We now explicitly state that **Tables 1–3 all use the same unified model trained once**. These tables are intended to evaluate whether one model can remain robust under diverse distortions and sampling-rate configurations—not to claim SOTA in each isolated task through task-specific tuning.

---

## **(2) Expanded Evaluation on Additional Datasets**

We substantially broadened our evaluation following reviewer suggestions:

- **VoxCeleb**
- **LibriTTS test-other**
- **URGENT 2025 blind test set**
- **DNS 2020**

Consolidated in Table 4, these results show that TF-Restormer generalizes well across **diverse corpora** despite being trained only on VCTK. Results on DNS 2020 is included in Appendix and anaylzed comprehensively in terms of both signal fildeity and perceptual quality, showing clear trade-off between two aspects.

**In addition, to ensure fairness, we added dedicated and standardized evaluations:**

- **Dedicated TF-Restormer variants** trained with task-matched pipelines (DN, SSR). These outperform the unified model, showing that unified training is strictly more general and not advantaged by data design.
- **URGENT-style training and official evaluation**, providing a fully standardized benchmark.

    Note: URGENT focuses on **matched-rate enhancement** and tends to favor **deterministic and large-capacity systems**, though not required. TF-Restormer is oriented toward **arbitrary-rate reconstructive restoration**, so its URGENT performance is naturally lower than top-ranked systems.


Together, these expanded experiments provide a fuller assessment of both **cross-corpus generalization** and **standardized matched-rate performance**, while highlighting the distinction between TF-Restormer’s intended scope and the constraints of URGENT.

---

## **(3) Strengthened Ablations**

To address concerns regarding key components, we expanded the ablations substantially:

- **Scaled log-spectral loss:** We now include fixed-scale variants, show that plain log1p behaves similarly to ℓ₁, and demonstrate that using constant weight values requires tricky tuning and **adaptive scaling** provides the most stable gradients and largest robustness gains under severe distortions. We also validate the effectiveness of perceptual loss
- **Shared SFI-STFT discriminator:** Comparing with rate-specific discriminators shows that the shared version yields more coherent supervision across sampling rates and higher overall quality.
- **Decoder design:** We clarify the model size and computational cost, and additionally evaluate a reduced version of the proposed model by reducing the encoder size to demonstrate that its effectiveness does not stem from parameter count or computational advantages.
- **Frequency projection module:** Both **shared** and **non-shared** variants are evaluated; the shared form offers the best balance of performance, stability, and parameter efficiency.
- **Undertraining of extension queries:** Ablations varying input/output rate distributions confirm robustness except under extreme (<1%) imbalance, with ~10% exposure sufficient to eliminate the issue.

These analyses strengthen the rationale behind each module and clarify their contributions to robustness.

---

## **(4) Closing Remark**

We believe the expanded evaluations, clarified intentions, and strengthened ablations fully address the reviewers’ concerns. The revision incorporates all major suggestions—additional datasets, standardized benchmarks, clearer architectural motivation, and more comprehensive analysis. We thank the reviewers again for their insightful feedback.

---

### Meta-Review · Area_Chair_aNy6 · 2025-12-26

**Summary:**

In this paper, the exclusive reliance on small-scale training data fails to establish a fair comparison against baselines trained on standard large-scale corpora and undermines the credibility of the claimed performance gains. The architecture represents an incremental engineering assembly of established modules rather than a methodological breakthrough. The necessity of the complex asymmetric design and shared discriminator remains unproven against simpler alternatives rendering the practical utility of the framework unsubstantiated.

**Reviewer Concerns:**

The concern regarding the limited architectural novelty of assembling established modules remains unresolved. The claimed superiority of the shared discriminator strategy over simpler independent baselines lacks definitive empirical justification. Furthermore, the issue of experimental unfairness stemming from mismatched training data scales and the insufficient theoretical grounding for the proposed loss function persist.

**Reviewer Scores:**

Reviewer NvNb might maintain the high score following satisfaction with the added benchmarks and ablations.

Reviewers 5YCi, 7rbh and HvyR would likely persist with their rejection ratings because the core objections regarding experimental unfairness and limited architectural novelty remain unaddressed.

---

### Decision · Program_Chairs · 2026-01-26

Reject